# Exocyst dynamics during vesicle tethering and fusion

Syed Mukhtar Ahmed ⬡ [1], Hisayo Nishida-Fukuda[1,2,3,7], Yuchong Li[4,5], W. Hayes McDonald[6], Claudiu C. Gradinaru[4,5] & Ian G. Macara[1]

The exocyst is a conserved octameric complex that tethers exocytic vesicles to the plasma membrane prior to fusion. Exocyst assembly and delivery mechanisms remain unclear, especially in mammalian cells. Here we tagged multiple endogenous exocyst subunits with sfGFP or Halo using Cas9 gene-editing, to create single and double knock-in lines of mammary epithelial cells, and interrogated exocyst dynamics by high-speed imaging and correlation spectroscopy. We discovered that mammalian exocyst is comprised of tetrameric subcomplexes that can associate independently with vesicles and plasma membrane and are in dynamic equilibrium with octamer and monomers. Membrane arrival times are similar for subunits and vesicles, but with a small delay (~80msec) between subcomplexes. Departure of SEC3 occurs prior to fusion, whereas other subunits depart just after fusion. About 9 exocyst complexes are associated per vesicle. These data reveal the mammalian exocyst as a remarkably dynamic two-part complex and provide important insights into assembly/disassembly mechanisms.

[1] Department of Cell and Developmental Biology, Vanderbilt University School of Medicine, Nashville, TN 37240, USA. [2] Department of Biochemistry and Molecular Genetics, Ehime University Graduate School of Medicine, Toon, Ehime 7910295, Japan. [3] Department of Hepato-Biliary-Pancreatic and Breast Surgery, Ehime University Graduate School of Medicine, Toon, Ehime 7910295, Japan. [4] Department of Physics, University of Toronto, Toronto, ON M5S 1A7, Canada. [5] Department of Chemical & Physical Sciences, University of Toronto Mississauga, Mississauga, ON L5L 1C6, Canada. [6] Department of Biochemistry, Vanderbilt University School of Medicine, Nashville, TN 37240, USA. [7] Present address: Department of Genome Editing, Institute of Biomedical Sciences, Kansai Medical University, Hirakata 5731010, Japan. These authors contributed equally: Syed Mukhtar Ahmed, Hisayo Nishida-Fukuda. Correspondence and requests for materials should be addressed to S.M.A. (email: syed.m.ahmed@vanderbilt.edu) or to I.G.M. (email: ian.g.macara@vanderbilt.edu)

Traffic between membrane-bound compartments requires the docking of cargo vesicles at target membranes, and their subsequent fusion through the interactions of SNARE proteins. The capture and fusion of vesicles are both promoted by molecular tethers known as multisubunit tethering complexes[1]. One group of such tethers, sometimes called CATCHR (complexes associate with tethering containing helical rods) comprises multisubunit complexes required for fusion in the secretory pathway, and includes COG, Dsl1p, GARP, and the exocyst[2]. The endolysosomal pathway contains two different tethering complexes, CORVET and HOPS, with similar overall structures to the CATCHR group[3].

COG consists of two subcomplexes, each containing four subunits, which function together within the Golgi[4–6]. The exocyst is also octameric, and is necessary for exocytic vesicle fusion to the plasma membrane (PM), but the organization of the complex has been controversial[7–10]. Several studies in yeast suggest that one (Sec3) or two (Sec3 and Exo70) subunits associate with the PM and recruit a vesicle-bound subcomplex of the other subunits, but other work argues that the exocyst consists of two subcomplexes of four subunits each that form a stable octamer or, in mammalian cells, that fivesubunits at the PM recruit three other subunits on the vesicle[11–22]. Rab GTPases promote exocyst binding to the vesicle, and SNARES, Rho family GTPases, the PAR3 polarity protein, and phosphoinositide-binding domains are all involved in recruiting an exocyst to the PM[20,23–30].

Despite advances in structural studies, we still know very little about how an exocyst functions. The dynamics, location, and regulation of exocyst assembly and disassembly remain unresolved. In mammalian cells, the overexpression of individual exocyst subunits causes aggregation and degradation[31]. A pioneering approach to avoid this problem involved silencing the Sec8 subunit and replacement by a Sec8-RFP fusion[31]. Sec8-RFP arrival at the PM was tracked using total internal reflection microscopy (TIRFM), which occurred simultaneously with vesicles ~7.5 s prior to vesicle fusion[31]. However, the behavior of other exocyst subunits was not addressed. In budding yeast, vesicles remain tethered for about 18 s prior to fusion, and several exocyst subunits were shown to depart simultaneously at the time of fusion, suggesting that the complex does not disassemble[21]. However, the time resolution was only ~1 s, so rapid dynamics could not be tracked.

The advent of CRISPR/Cas9-mediated gene editing coupled with the development of high-efficiency scientific CMOS (sCMOS) cameras has the potential to revolutionize our understanding of protein dynamics in the living cell. We have exploited these technologies to generate multiple tagged alleles of exocyst subunits by gene editing, and coupled proteomics with high-speed TIRFM and fluorescence cross-correlation spectroscopy (FCCS) to quantify exocyst dynamics in unprecedented detail. We discovered that, in mammary epithelial cells, exocyst connectivity is different from previous models of the mammalian exocyst but is consistent with the proposed connectivity in budding yeast[19], with two tetrameric subcomplexes, SC1 and SC2, that associate to form the complete octamer. Unexpectedly, each subcomplex can associate with the PM independently of the other, but both are required for vesicle docking. Subunit arrival at the PM coincides with vesicle arrival, but with a bias toward the prior arrival of SC2, which contains Exo70. Moreover, one subunit, SEC3, which is part of SC1, preferentially departs before fusion and the departure of other subunits, and exhibits anomalous diffusion. Cross-correlation of SEC3 to other subunits is significantly reduced. Taken together, these data are inconsistent with prior exocyst models and suggest that, in mammalian cells, exocyst subunits are in dynamic equilibrium with assembled complexes and the PM, that intact subcomplexes assemble on secretory vesicles as they dock, and that SEC3 is preferentially released prior to fusion.

## Results

**Generation of endogenously tagged exocyst subunits.** Each of the eight exocyst subunits can be C-terminally tagged in *S. cerevisiae* without disrupting function[19]. Therefore, we attempted to incorporate superfolder (sf) GFP, mScarlet-i (Sc), or Halo tags in-frame C-terminally into all exocyst subunit genes of the NMuMG murine mammary epithelial cell line, using Cas9-mediated gene editing (Fig. 1a and Supplementary Fig. 1a). Successfully gene-edited cells were isolated by FACS, and single colonies of cells were expanded and genotyped (Fig. 1b and Supplementary Fig. 1b–h). GFP+cell lines were recovered with five of the eight targeting vectors (Fig. 1c). For the other subunits, either no GFP+cells were detected (EXO84), or rare single GFP+cells were detectable after transfection (Supplementary Fig. 1i,j) but did not proliferate (SEC10, SEC15). Tagging these subunits at the N terminus was also unsuccessful. NMuMG cells are unusually sensitive to defects in membrane protein delivery caused by silencing of exocyst subunit expression, and rapidly apoptose[11,24,30,32–35]. Therefore, we conclude that tagging SEC10, SEC15, or EXO84 disrupts exocyst function, while C-terminal tags on the other five subunits (EXO70, SEC3, SEC5, SEC6, and SEC8) are well-tolerated. Homozygous clones of EXO70-GFP, SEC5-GFP, and SEC8-GFP were isolated, plus heterozygous clones of SEC3-GFP and SEC6-GFP (Fig. 1c, and Supplementary Fig. 1b–e, g). We also created homozygous double knock-in cell lines of SEC5-Sc or –Halo + EXO70-GFP (Supplementary Fig. 1h).

Significant GFP fluorescence above the autofluorescence background was detectable by confocal microscopy of live cells for each of the five single knock-in lines (Fig. 1d). SEC5, SEC6, SEC8, and EXO70 were enriched at intercellular junctions, consistent with data on fixed and immunostained epithelial cells, but the knock-in cells also have substantial diffuse cytoplasmic pools, and no detectable nuclear signal, contrary to some other reports[36]. SEC3 showed a predominantly cytoplasmic distribution with no significant accumulation at junctions.

To further validate the functionality of the tagged exocyst subunits, we performed immunoprecipitations from EXO70-GFP, SEC3-GFP, and SEC5-GFP cell lysates using GFP-trap beads, and analyzed them by mass spectrometry (MS). Each tagged protein co-precipitated all other seven subunits (Fig. 1e, and Supplementary Table 1) plus known regulatory factors (the small GTPases RALA and RALB), RHOGEFs, and proteins associated with intercellular junctions, but no RAB or RHO family GTPases[22,26,37–39]. We also identified several peptides from a SEC15-like gene, SEC15L. However, no peptides from four SEC6-related genes were detected, suggesting that these isoforms are not expressed in NMuMG cells. Independently, SEC8 binding to EXO70-GFP and SEC5-GFP was detected by immunoblots of GFP-trap bead precipitations (Supplementary Fig. 1k).

Interestingly, we also identified the SNARE-associated protein SNAP23 binding to EXO70-GFP, by immunoblot and MS of co-immunoprecipitations (Supplementary Fig. 1k, and Supplementary Table 1). However, no SNAP23 association with SEC5-GFP was detectable, suggesting that SNAP23 is likely not associated with the entire exocyst complex. Additionally, we did not detect any SEC1 or other SM proteins. Finally, the tagged cell lines showed similar proliferation rates to one another (Supplementary Fig. 3a). We conclude that the C-terminally tagged subunits retain normal protein–protein interactions and are biologically functional.

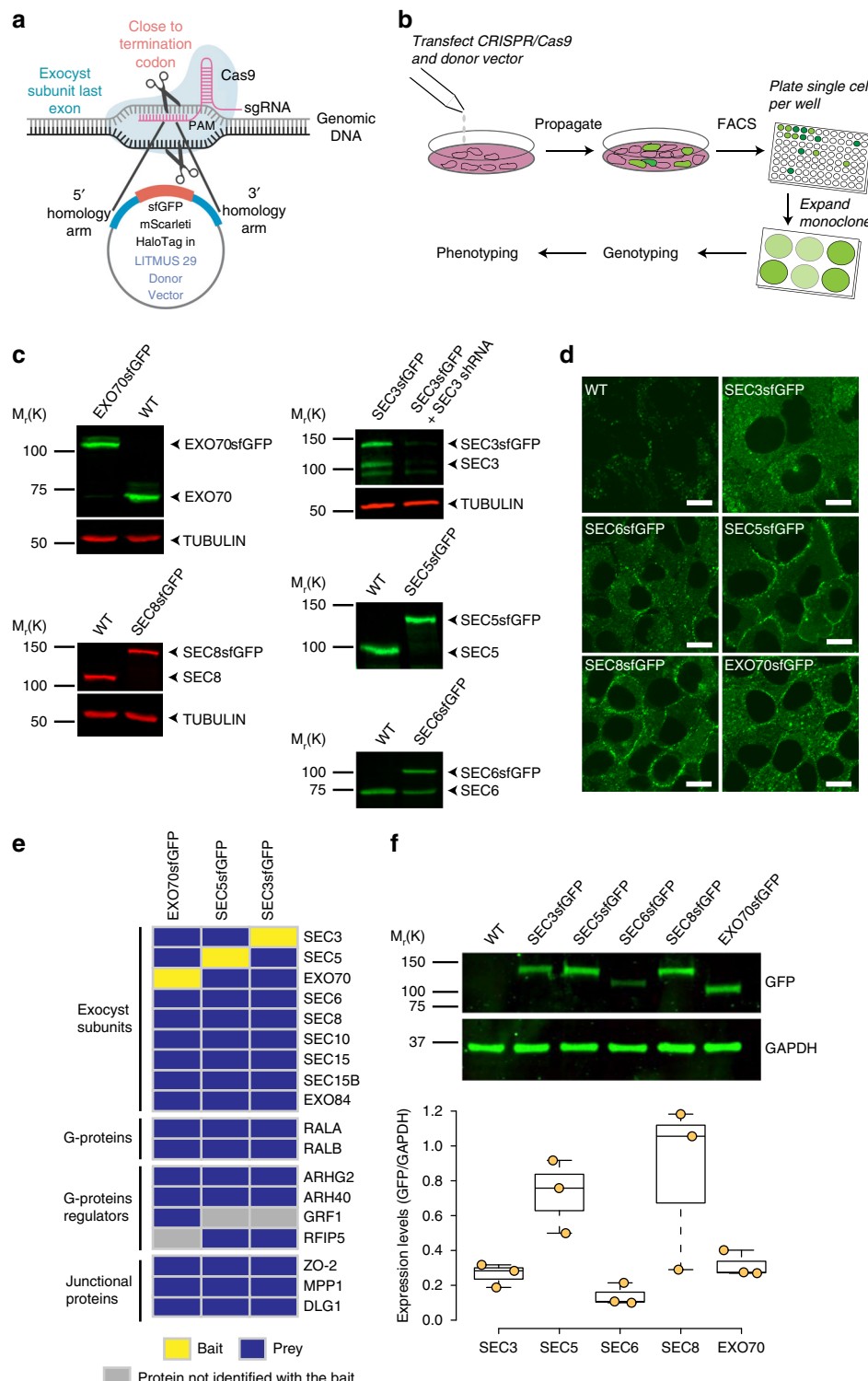

**Fig. 1** Establishment and validation of tagged exocyst subunit cell lines. **a** Schematic showing a region of subunit gene targeted by sgRNA and tags to be inserted C-terminally, in-frame with the coding region. **b** Strategy to isolate CRISPR/Cas9-mediated sfGFP-tagged clones of exocyst subunits in NMuMG cells. **c** Western blots with subunit specific antibodies show successful incorporation of sfGFP in both alleles (EXO70, SEC5, and SEC8), or in one allele (SEC3 and SEC6). A shRNA specific to SEC3 was used to confirm the identity of multiple bands in the blot. **d** Confocal images of live wild-type NMuMG cells and endogenously tagged SEC3-GFP, SEC5-GFP, SEC6-GFP, SEC8-GFP, and EXO70-GFP cell lines. Scale bar = 10 μm. **e** Protein–protein interaction heat map for endogenous SEC3-GFP, SEC5-GFP, and EXO70-GFP pulldown using GFP-Trap nanobodies and MS. Yellow indicates baits, blue the protein identified with high confidence, and gray denotes preys that were not identified. The experiments were repeated three times for EXO70-GFP and twice for SEC3-GFP and SEC5-GFP. All experiments were also partially confirmed by IP–WB in at least three independent experiments, with similar results. **f** Western blot analysis to assess the relative abundance of exocyst subunits fused to sfGFP, using anti-GFP antibodies (SEC3 and SEC6 quantifications corrected for heterozygosity). GAPDH was the loading control. Y axis shows the ratio of GFP/GAPDH. Quantification shows pooled data from three experiments

We took advantage of the sfGFP-tag to compare relative expression levels of individual exocyst subunits in NMuMG cells. Surprisingly, as shown in Fig. 1f, these levels are not equimolar. In particular, SEC5 and SEC8 are expressed at super-stoichiometric levels compared with the other three subunits (corrected where appropriate for heterozygosity). SEC8-GFP in particular was approximately threefold higher than expected. One interpretation is that addition of the sfGFP-tag stabilizes these proteins. However, blotting of cell lysates for SEC8 from two heterozygous SEC8-GFP clones showed that untagged and tagged alleles are expressed at the same levels (Supplementary Fig. 1g). These data suggest that in addition to forming exocyst complexes, a surplus of free SEC8 and SEC5 is present in cells. Whether they participate in other protein–protein interactions or have independent functions remains to be investigated.

**Connectivity of the mammalian exocyst**. It has been proposed that exocyst in budding yeast is a stable octamer comprised of two subcomplexes (SC1: SEC3, SEC5, SEC6, and SEC8, and SC2: EXO70, EXO84, SEC10, and SEC15)[19,40]. To investigate subunit connectivity in the mammalian exocyst, we silenced SEC8 in Exo70-GFP cells using a previously validated hairpin RNA[30], then captured the complex on GFP-trap beads, and performed a quantitative full-scan LC–MS/MS. Experiments were performed 3 d post transduction of shRNAs, before apoptosis occurs, and loss of SEC8 was confirmed by immunoblot analysis. Interestingly, only proteins corresponding to SC2 were recovered; those from SC1 were absent (Fig. 2a and Supplementary Fig. 2a, b). Loss of SEC8 also strongly reduced detection of both RALA and RALB, which bind not only to SEC5 in SC1 but also to EXO84 in SC2 (Supplementary Fig. 2b).

Conversely, we silenced SEC10 expression[30] by RNAi and captured SEC3-GFP, and then subjected the precipitate to immunoblotting for SEC6, SEC8, EXO70, and EXO84 (Fig. 2b and Supplementary Fig. 2h, i). In this experiment, components of SC1 (SEC6, SEC8) co-precipitated with SEC3, but not those of SC2 (EXO70, EXO84). Importantly, however, silencing of SEC10 did not result in the loss of SC2 components, suggesting that the integrity of the octameric complex or subcomplexes is not required for subunit stability (Supplementary Fig. 2h, i). Together, these data argue that the mammalian exocyst connectivity is similar to that in yeast, with two tetrameric subcomplexes. Moreover, each subcomplex needs to be intact for interaction with the other, and loss of one subunit disrupts the integrity only of its cognate subcomplex.

The functionality of the subcomplexes has never been addressed. To explore this issue, we asked if they can associate with the PM independently or only as an octameric complex. Association with basal PM was assessed by TIRFM. Subconfluent cells were used because vesicle fusion frequency at the basal membrane decreased substantially with confluency, when vesicles are redirected to apical junctions[41]. SEC3- and SEC8-GFP (SC1) and EXO70-GFP (SC2) were detected as fluorescent puncta in the unperturbed cell lines, with ~70% EXO70-GFP and SEC8-GFP associated with vesicles (Supplementary Fig. 2l–n). Unexpectedly, however, when SEC10 (in SC2) was silenced, SEC3-GFP remained detectable on the basal membrane, although EXO70-GFP was lost (Fig. 2c, d). SEC8 association with vesicles was slightly reduced (to ~45%; Supplementary Fig. 2m, n). A similar experiment using SEC5-GFP cells showed no impact of silencing SEC10 on the association of SEC5 with the membrane (Fig. 2e).

Conversely, silencing of SEC8 or SEC3 (in SC1; Supplementary Fig. 2c) caused depletion of SEC5 (SC1) but not EXO70 (SC2) from the membrane (Fig. 2f, g, and Supplementary Fig. 2j). Consistent results were also obtained when EXO70 was deleted by

Cas9-mediated gene editing: SEC8 (in SC1) was retained at the membrane (Supplementary Fig. 2f,g, k). From these data, we conclude that each subcomplex remains intact in the absence of its partner subcomplex, and, unexpectedly, retains association both with the PM and with vesicles.

**Exocyst dynamics during vesicle delivery and fusion**. Do all exocyst subunits arrive at the PM together with an exocytic vesicle, or do the subcomplexes or individual subunits arrive separately? To address this question, we transfected NMuMG cells with vectors to express mApple-Rab11 as a vesicle marker plus VAMP2-pHluorin, a vesicle-targeted pH-sensitive GFP that becomes highly fluorescent upon vesicle fusion with the PM. Arrival of Rab11+ vesicles at the membrane can then be correlated with the fusion event by two-channel time-lapse TIRFM (Fig. 3a). A typical kymograph is shown in Fig. 3b, and analysis of multiple events provided a median value of −14.5 s for arrival (Fig. 3c).

We next asked if vesicle fusion requires an exocyst, from measurements of events/minute/field in cells separately depleted of each of five subunits (Supplementary Fig. 2c–g). Because silencing of EXO70 by shRNA was not efficient, we used gRNA-targeted gene disruption for this subunit (Supplementary Fig. 2f, g). Importantly, in each case, loss of expression of subunits from either SC1 or SC2 caused a significant reduction in fusion events (Fig. 3e). We conclude that although the subcomplexes are competent to associate independently with the PM, they cannot separately promote vesicle fusion.

Based on this foundation, we simultaneously tracked exocyst subunit arrival and vesicle fusion, using a transferrin receptor linked to pHuji, a pH-sensitive RFP (TfR-pHuji). Importantly, this sensor marks the same vesicles as VAMP2-pHluorin (Fig. 3d). Representative kymographs are shown in Fig. 3f, and data are quantified in Fig. 3g. Interestingly, the four tested subunits (SEC3, SEC5, SEC6, and EXO70) all arrive with median times of 10–14 s prior to fusion, which is close to the arrival time of the Rab11+ vesicles. This concordance is similar to that observed in budding yeast[21], and suggests that the exocyst arrives at the PM with a vesicle, rather than being pre-assembled on the membrane.

We next asked when exocyst subunits disappear from fusion sites. SEC5, SEC6, and EXO70 each depart at a median time of ~1.3 s post fusion. Unexpectedly, however (Fig. 3h), SEC3 was anomalous with a median departure time of 0.4 s prior to fusion, significantly different from the other subunits. This result is consistent with the reduced enrichment of SEC3-GFP detected at intercellular junctions (Fig. 1d), and implicates SEC3 in a unique function associated with vesicle fusion. An alternative interpretation is that SEC3-GFP fusion might be only partially functional, with defective binding to the exocyst complex. We do not believe this is the case, however, because the SEC3-GFP cell line proliferates normally (Supplementary Fig. 3a), and the abundance of SEC5-Halo at the PM is identical in lines that express SEC3-GFP versus SEC8-GFP (Supplementary Fig. 3b). Furthermore, we did not observe any differences in subunit interactions whether we used heterozygous SEC3-GFP or a homozygous SEC3-Sc (Supplementary Fig. 3c, d). However, our ability to employ SEC3-Sc for imaging was limited by rapid photobleaching of this fluorophore.

**Coincidence measurements between exocyst subunits by TIRF**. To test whether subunit arrivals at the PM are truly coincident, we used double knock-in cell lines SEC8-GFP + SEC5-Halo or EXO70-GFP + SEC5-Halo and assessed their fluorescence intensity trajectories with high time resolution. SEC8-sfGFP and SEC5-Halo almost always arrived in the TIRF field together;

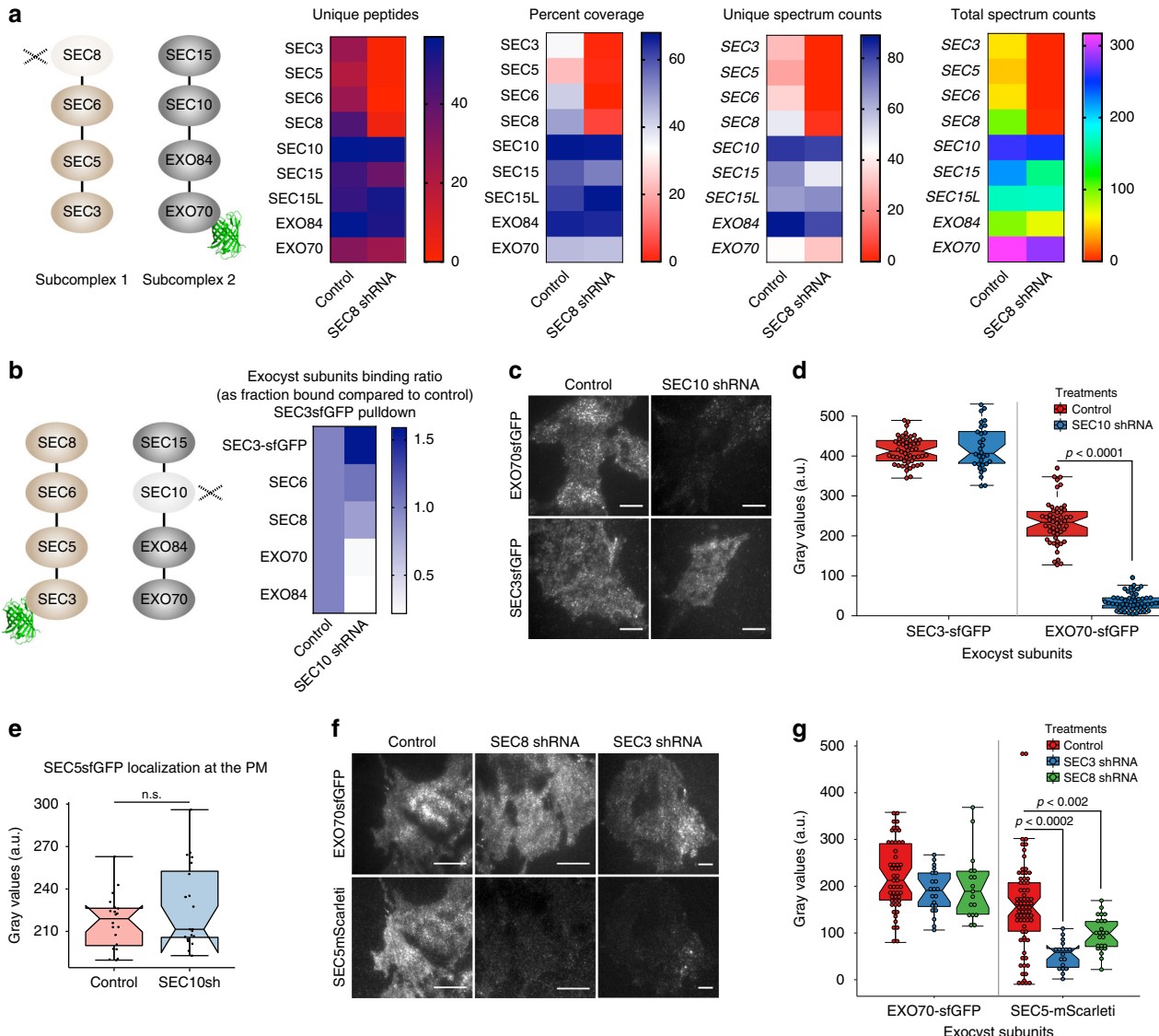

**Fig. 2** Mammalian exocyst complex comprises two subcomplexes that can localize to the PM independently. **a** Quantitative full-scan LC–MS/MS analysis of endogenous EXO70-GFP pulldowns from intact or Sec8-depleted cells. Inputs for the samples were calibrated and normalized using MRM–MS analysis on the same samples. The schematic shows the experimental design in which EXO70-GFP was captured using GFP-Trap beads and SEC8 was depleted by shRNA. **b** Heatmap summarizing relative binding of SEC6, SEC8, EXO70, and EXO84 to SEC3-GFP in SEC10-depleted cells compared to control shRNA-treated cells, as assessed by western blot. Also see Figure S2H-I. **c** TIRFM images of EXO70-GFP and SEC3-GFP in control or SEC10 shRNA-treated cells. Scale bars 20 μm. **d** Quantification of relative fluorescence intensities in C. **e** Quantification of fluorescence intensities from TIRFM images of SEC5-GFP cells treated with control or SEC10 shRNA. **f** TIRFM images of double knock-in NMuMG cells, showing localizations of EXO70-GFP and SEC5-Sc at the PM in control, SEC8 shRNA, and SEC3 shRNA-treated cells. Scale bars = 20 μm. **g** Quantification of relative fluorescence intensities in D. Center lines show the median; box limits indicate 25th and 75th percentiles; whiskers extend 1.5X the IQR from the 25th and 75th percentiles. Experiments were repeated at least three times with similar results

strikingly, however, EXO70-GFP showed a median delay of 77 ms with respect to SEC5-Halo (Fig. 4a, b). This small but significant lag between SC1 and SC2 subunits could not have been detected by comparing single subunit versus vesicle fusion arrival times, as described in Fig. 3 and in other studies[21]. Notably, there is a clear bias toward the prior arrival of SC2 (~54% of events). SC1 arrived first in only ~7% of events, while the remainder arrived simultaneously with SC2, within the time resolution of our camera (Fig. 4c) suggesting that, although SC1 and SC2 can bind independently to the PM (Fig. 2), there might also be co-operative interactions between them.

We also measured residency times for exocyst subunits on the PM. Both EXO70-GFP+SEC5-Halo and SEC8-GFP+SEC5-Halo

pairs could be traced together for ~12–14 s (Fig. 4d), and these durations were similar to that of vesicle residency at the PM as assessed using Rab11-mApple (Figs. 3c, 4e). These observations indicate that while association between SC1 subunits SEC5 and SEC8 is stable and they arrive together at the membrane, association between SC1 and SC2 is dynamic and they can arrive independently at (or near) the PM. Nevertheless, once at the PM, SC1 and 2 remain associated, with a residence time similar to that of the vesicles.

**Single-molecule quantification of subunit interactions.** Whether vesicles destined for the PM are associated with an intact exocyst or with only a subset of exocyst subunits remains

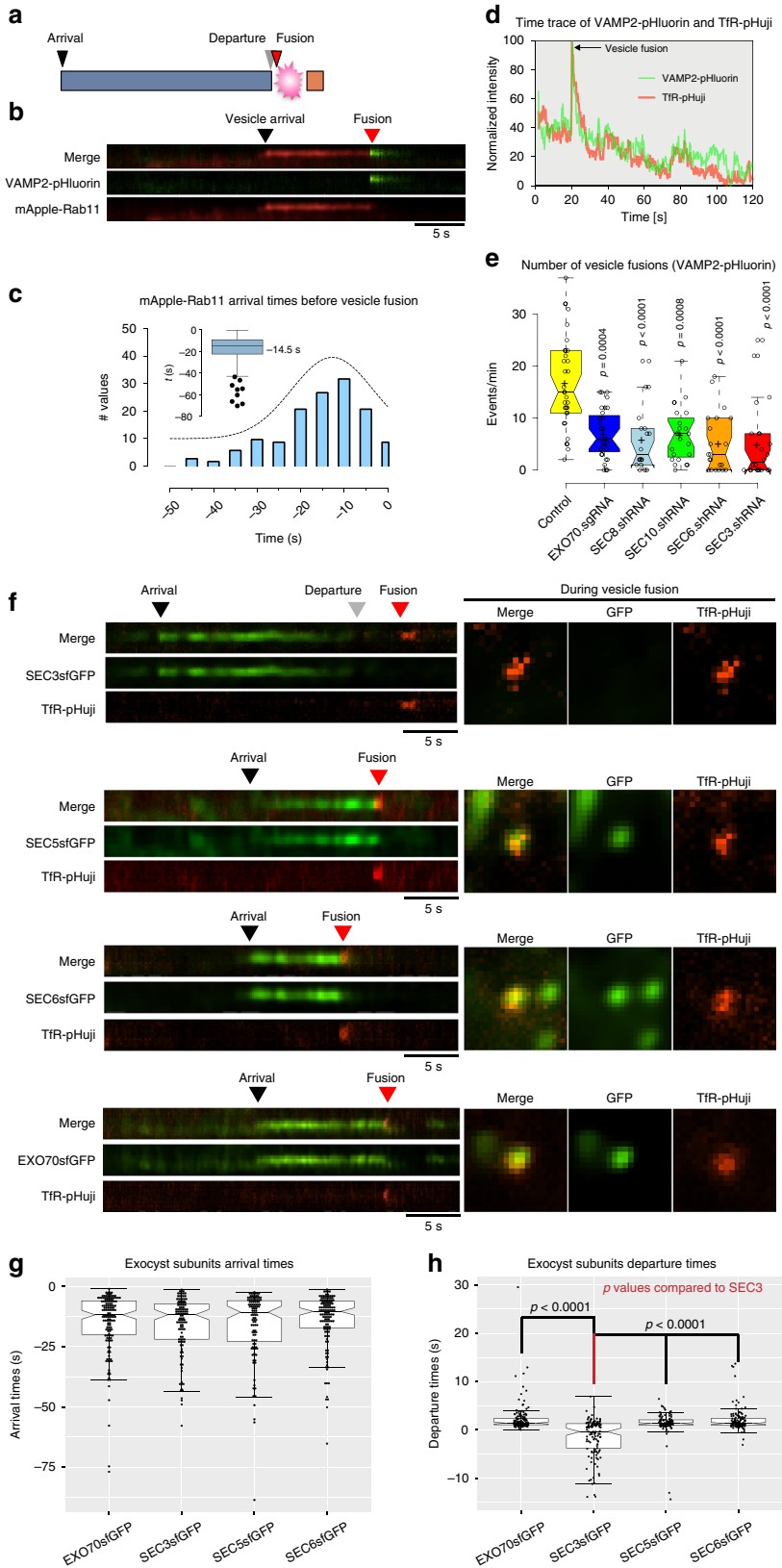

First, we needed to determine the stoichiometry of dye binding to Halo, as this parameter has not been previously reported. To

controversial[11–22]. To measure protein–protein interactions between subunit pairs throughout the cytoplasm, rather than just at the PM, we used single-molecule counting.

First, we needed to determine the stoichiometry of dye binding to Halo, as this parameter has not been previously reported. To

do so, we created a YFP-Halo fusion protein, expressed it in NMuMG cells (Supplementary Fig. 5a), and added Halo dye (J585-HTL) at different concentrations for various periods. Flow cytometry showed a strong linear correlation between YFP and dye fluorescence (Supplementary Fig. 5b). Next, cells were

**Fig. 3** Sec3 departs from the exocyst complex prior to vesicle fusion. **a** Schematic representation of the data. **b** NMuMG cells were transduced with mApple-Rab11 and VAMP2-pHluorin lentivirus and imaged by TIRFM. Kymographs show arrival and departure of mApple-Rab11 and vesicle fusion, highlighted by VAMP2-pHluorin flashes. Speed = 5 Hz. Scale bar = 5 s. **c** Distribution of mApple-Rab11 arrival times prior to vesicle fusions. The inset shows box and whisker plot; center line = −14.6 s (median; 95% CI: −16.6 to −12.5). Error bars = Tukey's range. **d** Intensity trace over time for VAMP2-pHluorin and TfR-pHuji. The peak intensities denote flashes resulting from vesicle fusion. Speed = 5 Hz. **e** Effects of exocyst subunits depletion on vesicle fusion activities in NMuMG cells assessed using VAMP2-pHluorin. $n = 32, 32, 18, 22, 21, 22$. P-values compared to control. **f** Kymographs showing an itinerary of SEC3-GFP, SEC5-GFP, SEC6-GFP, and EXO70-GFP arrivals/departures with respect to the vesicle fusion marker TfR-pHuji. Square images show snapshots in the green and red channels at the time of vesicle fusions. Scale bar = 5 s. **g, h** Quantification of exocyst subunit arrival and departure times at and from the vesicle fusion site shown in E. Data points shown as dots. **g** Median arrival times in seconds: −11.7 (95% CI: −12.9 to −9.1; EXO70), −11.6 (95% CI: −14.6 to −10.2; SEC3), −10.6 (95% CI: −14.0 to −8.2; SEC5), and −10.3 (95% CI: −12.3 to −8.8; SEC6). $n = 146, 132, 108$, and 132 objects in the order data shown. **h** Median departure times in seconds: 1.4 (95% CI: 1.2–1.5; EXO70), −0.4 (95% CI: −0.98 to −0.69; SEC3), 1.3 (95% CI: 1.21.6; SEC5), and 1.3 (95% CI: 1.21.7; SEC6). $n = 145, 117, 108$, and 132 particles from 33, 19, 29, and 26 cells in the order data are shown. Center lines = medians; hinges extend from 25th to 75th percentiles. Statistical significance measured by Kruskal–Wallis test followed by Dunn's post hoc analysis. Experiments were repeated three times with similar results

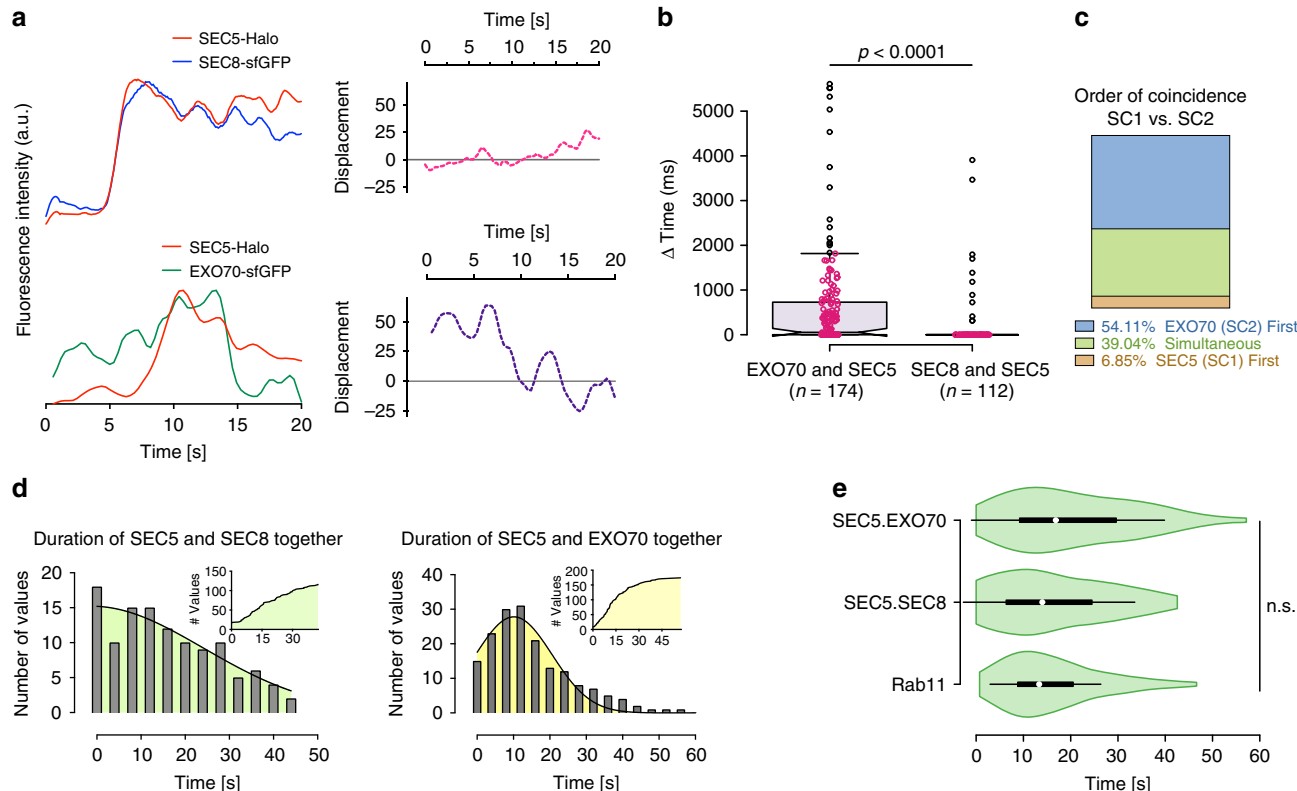

**Fig. 4** Coincidence measurements between exocyst subunit pairs by TIRF in live cells. **a** Fluorescence intensity trajectories (intensity values were normalized to a common scale) for the SEC5-Halo + JF585/SEC8-GFP pair, and SEC5-Halo + JF585/EXO70-GFP pair over 20 s. The fluorophores were excited simultaneously and images were captured at 12.5 Hz (left). (Right) Graphs show the displacement of the normalized intensities of the two fluorophores at each time point. **b** Quantification of the delay in coincidence between EXO70-GFP and SEC5-Halo + JF585 or SEC8-GFP and SEC5-Halo, measured by TIRFM. Outliers are represented by black circles; data points are plotted as magenta circles. The n denotes number of particles analyzed from a total of three experiments. **c** Distribution of the order of arrival of EXO70-GFP and SEC5-HaloJF585 shown in panels **a** and **b**. **d** Distributions of the duration of the indicated subunits were together in the TIRF field. **e** Comparison on the distribution of residence times of Rab11 and the indicated pairs of exocyst subunits. White circles = medians; box limits indicate the 25th−75th percentiles; whiskers extend 1.5X the interquartile ranges from 25th to 75th percentiles; polygons represent density estimates of data and extend to extreme values. Statistical significance measured by Kruskal–Wallis test followed by Dunn's post hoc analysis

ruptured in a small volume, which was cleared by centrifugation, and supernatants were spread onto coverglasses for single-particle counting by two-color TIRFM (Fig. 5a). Maximum fractional labeling efficiency was $0.40 \pm 0.014$ after 1.5 h, and increasing incubations to 18 h did not increase fractional labeling beyond 0.4 (Supplementary Fig. 5c, d). Additionally, FCCS measurements robustly detected a value of 0.4 for cross-correlation between Halo and YFP fluorescence in intact cells (Supplementary

Fig. 5e–h). Therefore, this value was used to correct our data on Halo-tagged subunit interactions.

Next, lysates from double knock-in cells expressing SEC5-Halo +SEC3-GFP, SEC8-GFP, or EXO70-GFP were rapidly spread on coverslips, and colocalization in single particles was quantified as described above (Fig. 5b). The majority of detected spots contained only single molecules of each species, as assessed by photobleaching, thus enabling a reliable assessment of fractional

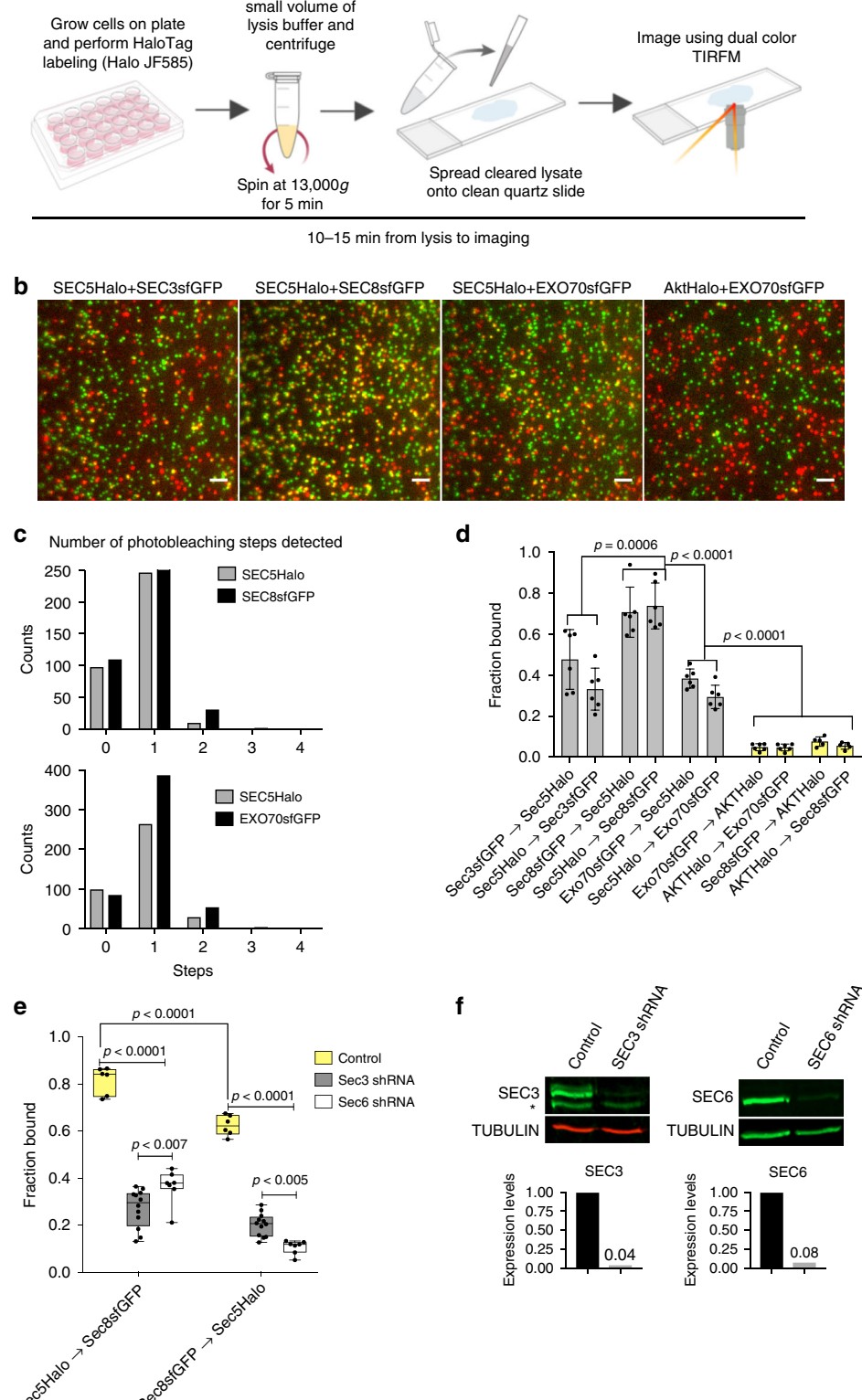

**Fig. 5** Assessment of fractional binding using single-molecule approach. **a** Diagram showing experimental workflow. **b** Cleared cell lysates from the indicated double knock-in cells were spread between quartz slides and glass coverslips for TIRF imaging. The GFP and Halo + JF585 fluorophores were excited simultaneously with 488-nm and 561-nm lasers at 12.5 Hz for 40 s. Scale bar = 1 μm. **c** Number of photobleaching steps detected for SEC5-Halo and SEC8-GFP in experiments shown in panel **b**. **d** Quantification of the fraction of subunit A bound to (–>) subunit B from experiments shown in panel **b**. Numbers of particles analyzed for each pair are shown in Supplementary Fig. 4c. **e** Fraction of SEC5-Halo and SEC8-GFP bound to each other in SEC3 or SEC6 shRNA-treated cells compared to control. Center line denotes mean. **f** Western blot analysis of SEC3 and SEC6 shRNA knockdown efficiencies in the experiment shown in **e**. Error bars denote ± SD. Statistical significance was computed by one-way (**d**) or two-way (**e**) ANOVA, followed by Tukey's multiple comparison tests. Experiments were repeated three times and data were pooled

inter-subunit binding (Fig. 5c, Supplementary Fig. 4d). SEC3 counts were corrected for heterozygosity. An Akt-Halo fusion, expressed in cells at a similar concentration to the endogenous EXO70-GFP fusion (Supplementary Fig. 4a–c), was used as a negative control to correct for chance colocalization. At the fluorophore concentrations present, overlap was only 5–8% (Fig. 5d).

Coincidence within the diffraction limit between SEC8-GFP and SEC5-Halo was much higher, at ~70% (Fig. 5d, Supplementary Fig. 4d). Surprisingly, however, only ~40% of SEC3- and EXO70-GFP molecules colocalized with SEC5-Halo (Fig. 5d). Conversely, colocalization of SEC5-Halo to SEC3-GFP or EXO70-GFP was even lower, at ~30%, likely because SEC5 is expressed at a significantly higher level than SEC3 or EXO70 (Fig. 1f). These results were corroborated by co-immunoprecipitation of SEC8-GFP with ~80% of total SEC6 (in SC1) but only ~30% of total EXO70 (in SC2) (Supplementary Fig. 4f, g). These values may be underestimates because of dissociation after cell lysis, but they suggest, nonetheless, that some subunits within a subcomplex (SEC5, SEC6, and SEC8 in SC1) interact quite stably with one another, but that SEC3 binding is substoichiometric, and similar to the weaker interaction between subcomplexes.

Because SEC3 binds to other SC1 subunits relatively weakly, we asked if it is required for subcomplex stability. SEC3 or SEC6 (as a positive control) were efficiently silenced, and SEC5/SEC8 interactions were measured by single-molecule counting. Surprisingly, both perturbations strongly reduced SEC5/SEC8 binding (Fig. 5e, f). These data argue that SEC3 is essential for the assembly of SC1 (and perhaps interaction with SC2), but not for maintenance of the SEC5/6/8 heterotrimer once it has formed. This idea is consistent with the early departure of SEC3 from the exocyst complex prior to vesicle fusion (Fig. 3f, h).

**FCCS suggests a fraction of the complex is preassembled.** As an independent approach to determine the fractional association of SC1 and SC2, we performed in-cell dual-color FCCS (Fig. 6a). We assessed co-diffusing fractions for EXO70-GFP, SEC8-GFP, or SEC3-GFP with SEC5-Halo labeled with JF-646-Halo ligand (Eq 3). The data were corrected for maximum detectable correlation based on the Venus(YFP)-Halo fusion protein, assuming that the fraction of fluorescent YFP is similar to that of sfGFP. About 92% of SEC8-GFP was coupled to SEC5-Halo within the cell and 85% near the bottom of the cell (Fig. 6c, e; Supplementary Fig. 6b, d). SEC3-GFP/SEC5-Halo coupling was ~66% within the cell and ~60% near the bottom (corrected for heterozygosity) (Fig. 6d, e; Supplementary Fig. 6c, d). We also estimated the association between SC1 and SC2 from the SEC5-Halo/EXO70-GFP pair, with ~65% of each of the cytosolic and membrane fractions co-diffusing (Fig. 6b) (Supplementary Fig. 6a, d). As an alternative approach for the membrane fraction, we measured colocalization within the TIRF field, which showed that ~85% of SEC5–SEC8 and ~60% of SEC5-EXO70 subunits are together (Fig. 6f), in close agreement with the FCCS data.

Data fitting (Eqs 2, 3, and 4), revealed two diffusion components, with the slower (49 ±10%) GFP-tagged population ($D = 0.54\ \mu m^2\ s^{-1} \pm 0.26$, approximating to a hydrodynamic radius of ~460 nm) within the cell being common among all the subunits and cross-correlating. The faster component, which does not cross-correlate, has a median diffusion rate of 14.5 $\mu m^2\ s^{-1}$, reminiscent of freely diffusing molecules in cells[42]. Importantly, EXO70-GFP, SEC8-GFP, and SEC3-GFP did not show any cross-correlation with the negative control, Akt-Halo (Fig. 6b–d). From these experiments, we conclude that about half to two-thirds of all exocyst subunits within the cytoplasm are octamers,

possibly associated with vesicles, while the remaining subunits are subcomplexes and a small proportion of freely diffusing monomers.

**Diffusivity of exocyst subunits at the plasma membrane.** The low abundance of SEC3-GFP at intercellular junctions and at the basal membrane, as compared with other subunits, suggests that this subunit has unique properties (Figs. 1d, 7a–b). These results, together with the observation that SEC3-GFP preferentially leaves the docked complex before the other subunits (Fig. 3f, h), suggest that SEC3 interacts unusually weakly with the complex. To further test this possibility, we measured subunit diffusivity at the PM by tracking individual particle trajectories, and calculated their mean squared displacements (Fig. 7c). SEC3-GFP diffusion was anomalous with a median of 0.12 $\mu m^2\ s^{-1}$ MSD compared with other subunits, which ranged from medians of 0.02–0.08 $\mu m^2\ s^{-1}$ (Fig. 7d–e). Total displacement measurements showed that SEC3-GFP motion was higher than that of other subunits, with a median of 0.21 $\mu m$ as compared with SEC8-GFP or EXO70-GFP (median 0.10 $\mu m$), and SEC5-Halo (median 0.07 $\mu m$) (Supplementary Fig. 7a). We conclude that SEC3 interacts with the complex with an unusually low affinity, suggesting that it might be the limiting factor in formation of an octamer.

**Quantification of exocyst molecules at docked vesicles.** To better understand how an exocyst tethers vesicles to the membrane, we quantified the numbers of complexes at tethered vesicles before fusion. Stepwise bleaching was not detected, suggesting that large numbers of molecules were present, so we used ratio comparison to fluorescent standards[43]. We fused one, two, or three sfGFPs in tandem to the C-terminus of CD86, a monomeric, PM-localized protein[44] and determined their photobleaching steps and initial intensities ($I_o$). These values were then used to generate a standard curve with a coefficient of determination >0.99 (Fig. 8a–b). To further ensure that we were measuring monomeric CD86, we immunostained fixed cells with anti-GFP antibodies conjugated to biotin and probed for photobleaching steps using streptavidin-ATTO488 (Supplementary Fig. 7b–d). These experiments confirmed that our constructs behave as expected. The empirically determined lateral point spread function (PSF) of our TIRF objective (N.A. 1.49) was 244 nm at a wavelength of 488 nm, and the size of the captured images was 120 nm per pixel (Supplementary Fig. 7h). To meet the Nyquist criterion, we limited our measurement of peak intensities to objects that were resolved by a distance of >4 pixels on the camera (Supplementary Fig. 7e–j).

To calculate the numbers of subunits at sites of vesicle fusion, we expressed the TfR-pHuji construct in the exocyst-GFP knock-in cell lines and measured the peak intensities of well-resolved particles of exocyst subunits at fusion sites (Fig. 8c–d). We estimate approximately nine molecules of each exocyst subunit associated with each vesicle during tethering (Fig. 8d), slightly less than that predicted for budding yeast[45].

## Discussion
A major goal of cell biology is to quantitatively assess the dynamics of molecular processes in living cells. This goal has often been confounded by perturbations introduced through overexpression of gene products, and by imaging limitations. In this study, we report the first quantitative analysis of endogenous exocyst dynamics in mammalian cells, at unprecedented resolution. We used CRISPR/Cas9-mediated gene editing to incorporate fluorescent tags at the C-termini of five exocyst subunits in NMuMG cells and assessed connectivity and abundance. We employed fast TIRF imaging of these endogenous molecules to

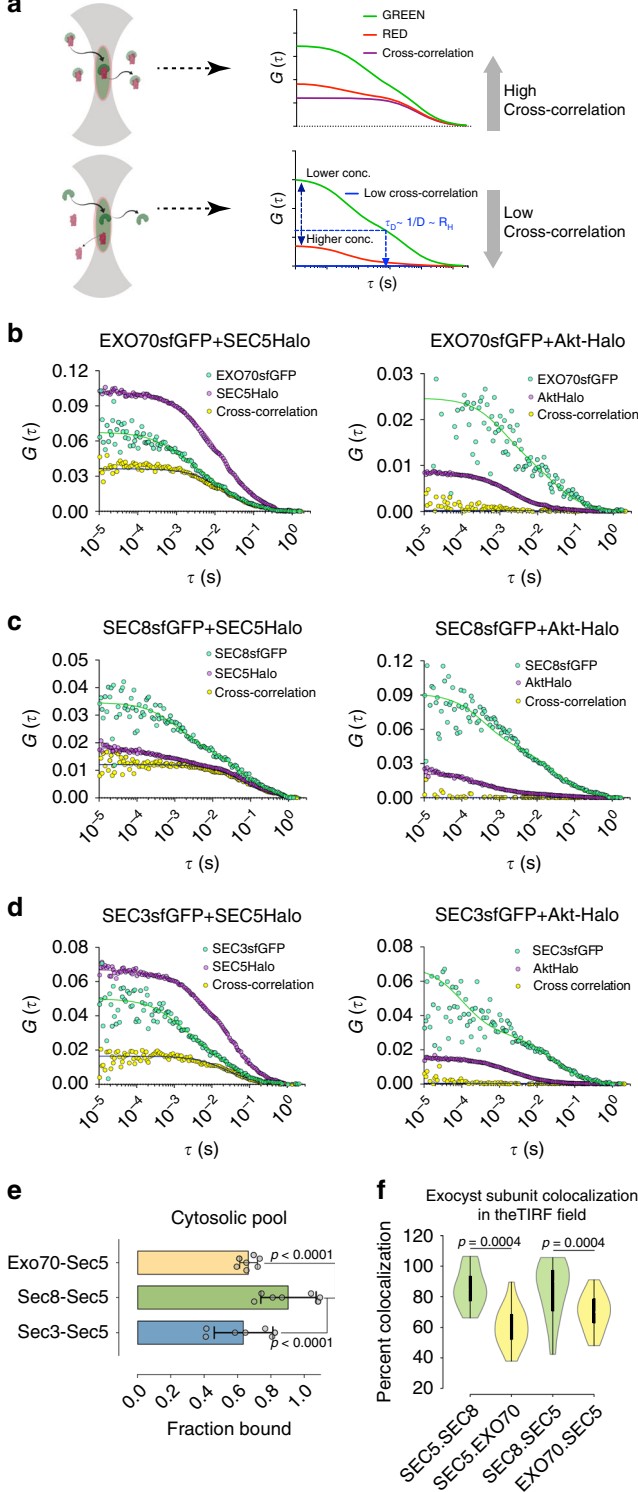

**Fig. 6** Dual-color FCCS of exocyst subunits in the membrane and cytosol. **a** Schematic of dual-color cross-correlation spectroscopy (FCCS). Top diagram shows cross-correlation between co-diffusing molecules. Bottom shows molecules that do not co-localize. **b** SEC5-Halo and EXO70-GFP FCS measurements in the cytosol and the PM. EXO70-GFP and Akt-Halo fluorescence cross-correlation was used as a negative control. HaloTag was labeled using JF646 Halo ligand (200 nM for 1.5 h). **c** SEC5-Halo and SEC8-GFP, or SEC8-GFP and Akt-Halo FCS measurements in the cytosol or the PM. **d** SEC5-Halo and SEC3-GFP, or SEC3-GFP and Akt-Halo FCS measurements in the cytosol or the PM. **e** Statistics of the fraction of GFP-tagged exocyst subunit bound in the cytosol. SEC8-GFP+SEC5-Halo: 92 ± 3.86%, SEC3-GFP + SEC5-Halo: 67 ± 3.84%, and EXO70-GFP+SEC5-Halo: 64 ± 3.46% (mean ± SEM). **f** Fractional coincidence between SEC5-Halo and SEC8-GFP or SEC5-Halo and EXO70-GFP was analyzed using TIRFM. Calculations were corrected for HaloTag labeling efficiency and expression levels of SEC8-GFP and EXO70-GFP with respect to SEC5-GFP. Center line denotes median. SEC5-Halo bound to SEC8-GFP was 85 ± 3.3% (mean ± SEM). SEC8-GFP bound to SEC5-Halo was 85 ± 5.1%. SEC5-Halo bound to EXO70-GFP was 60 ± 3.2% and EXO70-GFP bound to SEC5-Halo was 69.8 ± 2.9%. Object detection parameters were set between 0.30 and 0.35 μm and contrast was adjusted to faithfully detect objects of interest. White circles show the medians; box limits indicate the 25th and 75th percentiles as determined by R software; whiskers extend 1.5 times the interquartile range from the 25th and 75th percentiles; polygons represent density estimates of data and extend to extreme values. *p values* were calculated using Kruskal–Wallis test followed by Dunn's multiple comparison test. The hydrodynamic radius ($R_H$) for the slow diffusing fraction was median 466 nm (95% CI: 355.2–606.8), and that of the fast diffusing component, 14.82 nm (95% CI: 13.22–17.27). FCS measurements were fitted (lines) to the model described in Methods (Eqs 2–4). Curves shown are representative, taken from a single cell. Statistics are average from > 25 measurements for each condition from two experiments and > 5 cells. *p*-values were calculated by one-way ANOVA followed by Tukey's post hoc test

absent from the basal membrane, and is unlikely to be involved in recruitment to this region of the cell cortex. Indeed, when we expressed mApple-Par3 in the knock-in cell lines, no signal was detectable at the basal surface by TIRFM; and silencing of PAR3 expression did not reduce vesicle fusion frequency there (Supplementary Fig. 4h). Therefore, PAR3 is not necessary for basal vesicle tethering and fusion, and whether there are different exocyst receptors for different regions of the cell remains to be investigated.

The exocyst assembly mechanism remains controversial. In yeast, several lines of evidence suggest that exocyst is a stable octameric complex, comprised of two subcomplexes, that is bound to vesicles prior to their arrival at the PM[19,21]. This molecular organization is supported by the cryo-EM structure of the yeast exocyst and an earlier negative stain structure, both of which are consistent with an eight-component complex[19,40]. Interestingly, the structure of the yeast exocyst suggests a weaker affinity for SEC3[40], which is consistent with our observations, and with the fact that the yeast SEC3 is not essential for viability[46]. Others have proposed different models, in which SEC3 and/or EXO70 functions exclusively at the PM, and the remaining exocyst subunits arrive on secretory vesicles[17,18]. Moreover, there is still no consensus on whether exocyst binds to the PM prior to tethering incoming vesicles, or instead needs to bind a vesicle before associating with the PM[47].

In mammalian cells, five subunits on the PM were proposed to interact with three subunits on incoming vesicles[22]. On the other hand, a recent *bioRxiv* preprint reports that negative stain EM images of the mammalian exocyst expressed in insect cells closely

unravel arrival and departure times at vesicle fusion sites, and single-molecule approaches to estimate the fractional binding of exocyst subunits, and the number of subunit molecules per tethered vesicle. Together, these approaches enable a single-cell biochemistry that is broadly applicable, and in this study has provided comprehensive insights into mammalian exocyst behavior.

Previously, we reported that PAR3 interacts with exocyst in NMuMG cells and is required for membrane protein recruitment to lateral membranes and tight junctions[30]. However, PAR3 is

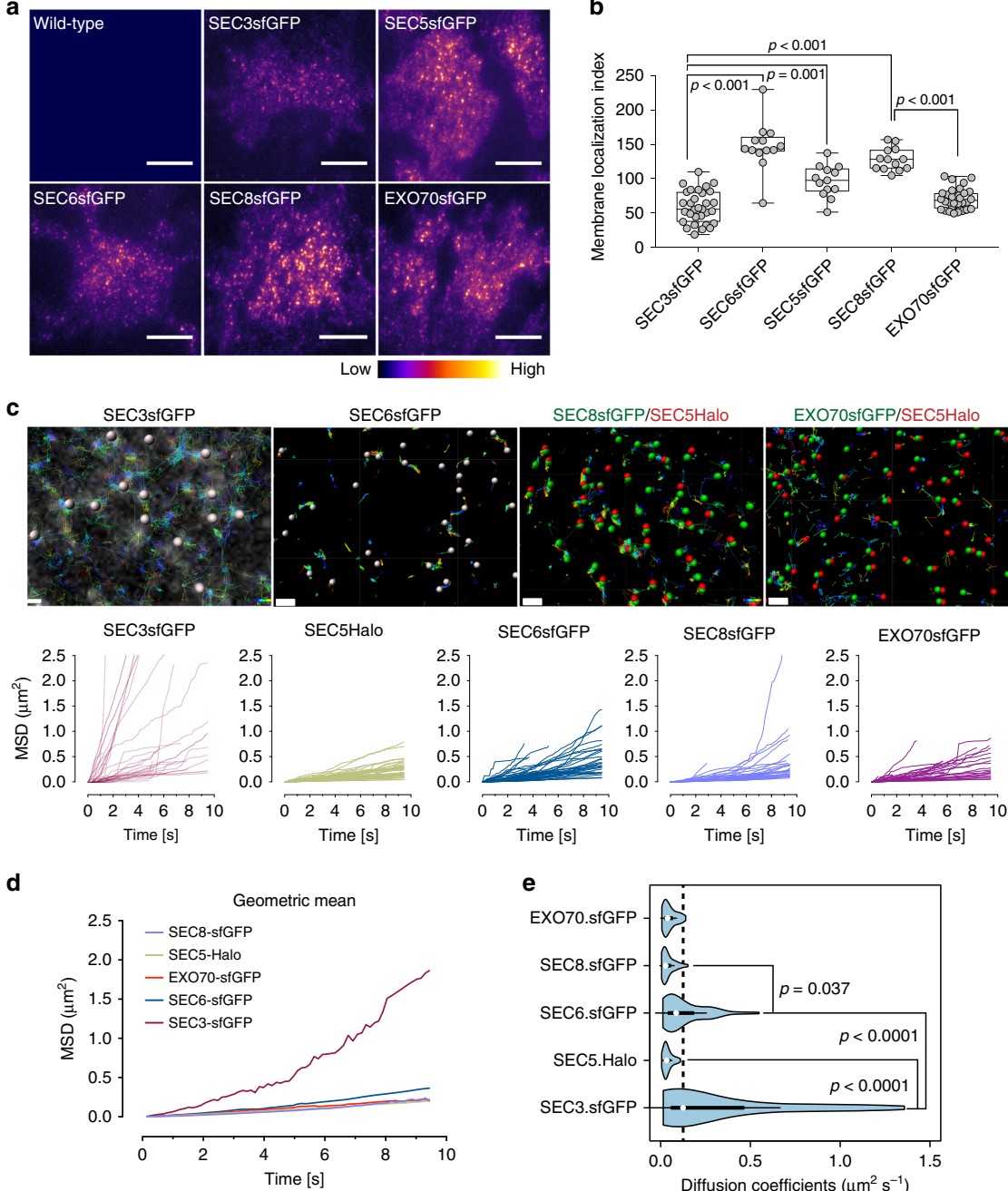

**Fig. 7** Measurements of diffusivity of exocyst subunits. **a** TIRFM images of untagged wild-type or indicated exocyst subunits fused to sfGFP (top). Heat map of above images (bottom). Scale bar = 20 μm. **b** Quantification of exocyst subunits localization at the TIRF field. Membrane localization index = density of spots * intensity of cells. Whiskers extend from min to max. Values for SEC3 and SEC6 were corrected for heterozygosity. n = 31, 13, 13, 13, 32 fields containing 2–4 cells each. **c** Particle tracking over time for the subunits indicated. For two-color tracking, each channel was tracked using Imaris software tracking algorithm and overlaid. Graphs show mean squared displacements over time. Scale bar = 0.5 μm (SEC3-GFP) and 0.8 μm (rest). **d** Representation of data in **b** as geometric means of MSDs < $r^2$ >. **e** Diffusion coefficients of the indicated subunits were measured from < $r^2$ >. Mean (μm² s⁻¹) ± SEM = 0.31 ± 0.05 (SEC3-GFP), 0.04 ± 0.002 (SEC5-Halo), 0.13 ± 0.01 (SEC6-GFP), 0.04 ± 0.004 (SEC8-GFP), and 0.05 ± 0.004 (EXO70-GFP). The effect size in terms of Cohen's d value between SEC3-GFP and SEC6-GFP is 0.348 and between SEC6-GFP and SEC8-GFP is 0.176. White circles show the medians; box limits indicate the 25th and 75th percentiles; whiskers extend 1.5 times the interquartile range from the 25th and 75th percentiles; polygons represent density estimates of data and extend to extreme values. Experiments were repeated at least three times with similar results. P-values were computed using one-way ANOVA test followed by Scheffe's multiple comparison tests

resemble the octameric yeast complex[48]. These data highlight the confusion surrounding the most basic organizational details of the exocyst. We have now discovered that, contrary to previous models, the mammalian exocyst is highly dynamic—at least within the living cell. Subunit connectivity is conserved between yeast and mammals, but unexpectedly the two subcomplexes can assemble and localize to the PM independently of each other, and associate with vesicles, yet neither is competent alone to promote vesicle fusion. Based on our corrected FCCS and TIRFM data, ~65% of SC1 and SC2 are associated in an octameric complex

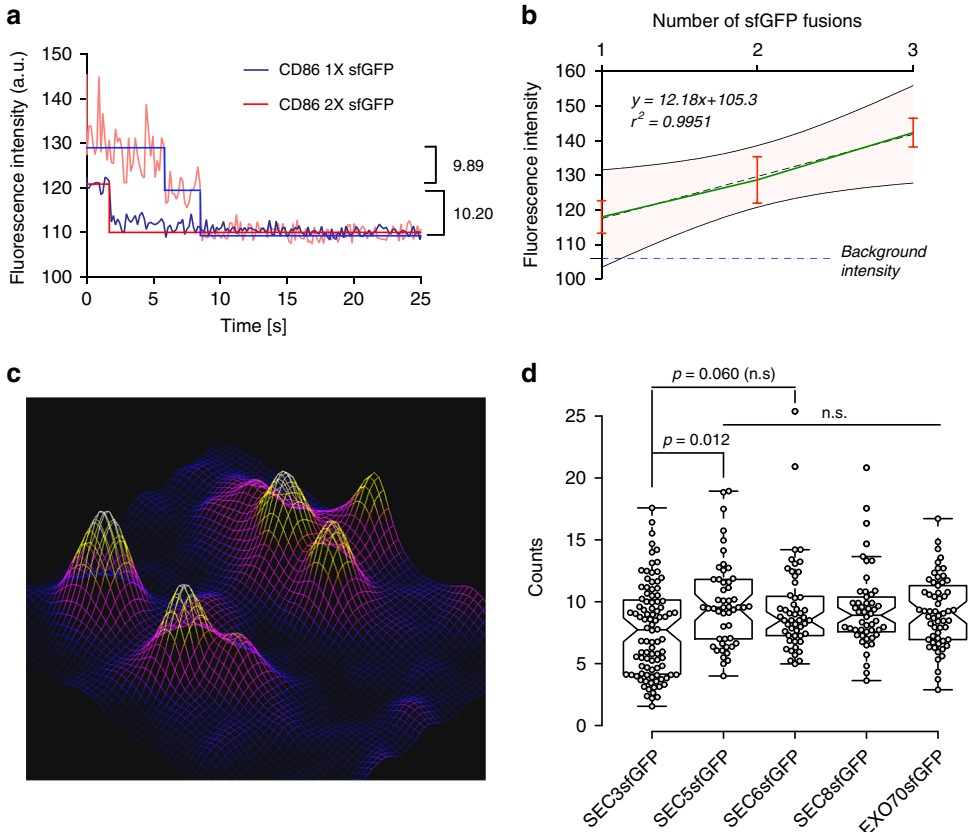

**Fig. 8** Exocyst molecule counting at vesicle fusion sites. **a** Stepwise photobleaching of CD86 fused to one or two sfGFP molecules used to determine intensities of known numbers of GFP molecules. **b** Standard curve for counting protein molecules, from intensities measured on CD86 fused to one, two, or three sfGFP molecules. Error bars = ±s.d.; pink shaded region denotes 95% confidence band. **c** Fluorescence landscape of SEC8-GFP shows a typical resolvable distribution of molecules within an ROI from which intensity measurements were taken. **d** Intensity (y) was measured from the peak and converted to the number of molecules using the regression equation determined in **b**. Numbers for SEC3 and SEC6 were corrected for heterozygosity. Mean ± SD = 7.6 ± 3.6 (SEC3-GFP), 9.8 ± 3.5 (SEC5-GFP), 9.8 ± 3.4 (SEC6-GFP), 9.35 ± 3.2 (SEC8-GFP), and 9.19 ± 2.8 (EXO70-GFP). Cohen's *d* value for the difference between SEC3-GFP and SEC5-GFP is −0.208. Centers indicate median, box limits indicate 25th and 75th percentiles, and whiskers extend 1.5x IQR from 25th to 75th percentiles. *n* = 96, 50, 51, 50, 58. Coefficient of variation (CV) = 47.9, 35.2, 39.2%, 34.3, and 30.7% in the order indicated in the graph. CV between each subunit was 7.9%. Experiments were repeated at least three times with similar results. *P*-values were computed using one-way ANOVA test followed by Scheffe's post hoc test

(Fig. 9a), while ~25% of SC1 exists as an independent sub-complex, and only about 10% of subunits are (presumably) free. Anomalously, SEC3 is relatively weakly associated with its sub-complex, SC1, with about 40% unbound from its subcomplex, suggesting that a significant fraction of SC1 might be a trimer rather than a tetramer (Fig. 9a). Together, these data indicate that the mammalian exocyst is in a dynamic equilibrium between free subunits, subcomplexes, and octamer, but that at steady state, about two-thirds of the subunits are assembled into the complete octameric complex.

Arrival times at the PM are similar for all exocyst subunits and vesicles, consistent with data from HeLa cells[31] and yeast[21], but we can in ~54% of events detect a small (median 80 msec) delay between SEC5 (on SC1) and EXO70 (on SC2), suggesting that octameric exocyst complex assembles on vesicles at or near the PM, perhaps from subcomplexes already attached to the vesicles. Importantly, there is a bias toward SC2 arriving prior to SC1 at the membrane, suggesting that although each can associate independently with the PM, there might also be co-operative interactions between them. Based on our data, we propose a comprehensive new model for exocyst interactions (Fig. 9) that

while consistent with previous observations expands our view of the dynamics and mechanism of this protein complex.

Finally, we found that in NMuMG cells, the SNARE protein SNAP23 interacts with EXO70 but not with SEC6 or other sub-units. No binding of syntaxins or SM proteins to any of the exocyst subunits was detectable. We speculate that mammalian EXO70 facilitates SNARE complex formation by bringing SNAP23 to the vesicle tethering site, and dissociation of SEC3 from the complex just prior to fusion then enables the SNARE complex to trigger membrane fusion (Fig. 9b–d). However, further studies using endogenously tagged SNARE proteins will be required to test this model. It will also be of great interest to track the dynamics of exocyst interaction with many other factors, including the RAL and RHO family GTPases, RABs, and myosins.

## Methods
**Plasmids constructs and other reagents**. pIRESpuro2-CD86-mEos2 and pIRESpuro2-CTLA4-mEos2 were gifts from Mike Heilemann (Addgene plasmids # 98284 and 98285). mEos2 was replaced with superfolder GFP coding sequence (sfGFP), which was PCR amplified from sfGFP-N1, a gift from Michael Davidson

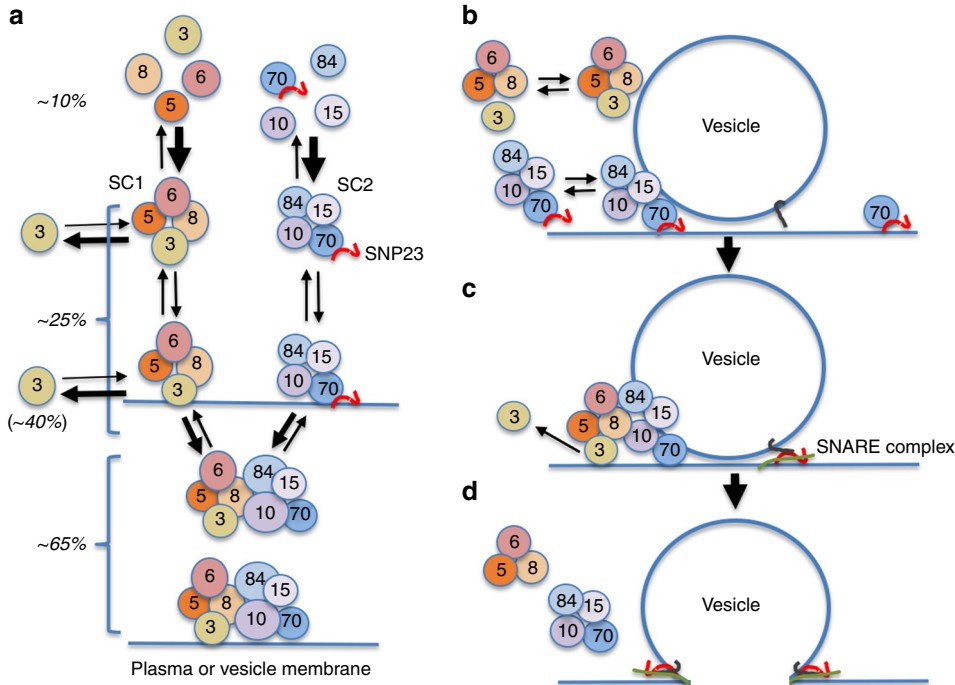

**Fig. 9** Model for exocyst subunit assembly/disassembly and vesicle tethering. **a** Schematic of subunit interactions within the cell. Subunits are shown as circles, SC1 and SC2 mark the two subcomplexes, and arrow weights represent relative on or off rates. SNP23 associates with EXO70 but is not co-precipitated with SEC5. Detailed molecular interactions with membranes are not illustrated and the diagram is not to scale. Percentages are the estimated proportions of each state, derived from corrected TIRFM and FCCS data. Octameric complex abundance = (SEC5 + EXO70) colocalization or cross-correlation; tetrameric subcomplex abundance = (SEC5 + SEC8)−(SEC5 + EXO70); and free subunit abundance = total−octamer−tetramer. Free SEC3 is calculated from total−(SEC3 + SEC5). We do not know what fraction of octamer is actually a heptamer lacking bound SEC3. FCCS data suggest that about 50% of exocyst is very slowly diffusing within the cell, likely associated with vesicles, estimated from calculating the hydrodynamic radius of the slow diffusing species. We propose that the individual subcomplexes can interact weakly with membranes but bind membrane more efficiently when assembled into an octamer. SEC3 interacts with SC1 less robustly than do the other subunits. **b** Individual subcomplexes can associate with vesicles at the PM but cannot trigger fusion. Percentages show estimated relative abundances at the membrane, based on TIRFM data. **c** Interaction of the subcomplexes to form octamer stabilizes tethering, and release of SEC3 precedes fusion. **d** Fusion is accompanied by rapid release of the exocyst

(Addgene plasmid # 54737), with flanking XmaI/NotI restriction enzyme sites and inserted at the AgeI/NotI site of the plasmid in frame with CD86. As such, the AgeI site at the beginning of sfGFP was destroyed and a new AgeI site created at the end of sfGFP before the stop codon, to enable insertion of subsequent tandem sfGFP coding sequences in frame. pEGFP-VAMP2 was a gift from Thierry Galli (Addgene plasmid # 42308). TfR-pHuji was a gift from David Perrais (Addgene plasmid # 61505). Supereclipliptic pHluorin (pHluorin2) was PCR amplified from VV063: 1xCox8-supereclipliptic pHluorin in fck, a gift from Adam Cohen (Addgene plasmid # 58500) to make VAMP2-pHluorin2. Halo-tag coding sequence was synthesized as a geneblock (IDT) to make VAMP2-Halo. Akt-Halo construct was generated by replacing mApple with HaloTag using the EcoRI/NdeI restriction sites at the C-terminal end of a pLVTHM-Akt-CA construct described previously[30]. For CRISPR/Cas9-mediated gene editing, locus-specific 5′ and 3′ homology arms were synthesized as gBlocks double-stranded gene fragments (IDT), or amplified by PCR. These gene fragments were designed to remove the stop codon from the gene ORF, delete the PAM sequence recognized by the cognate sgRNA, and to attach an in-frame tag (sfGFP, HaloTag, or mScarleti). They were then cloned into LIT-MUS29 vector together with the appropriate tag either using Gibson assembly (NEB) or by conventional cloning using the following restriction sites: KpnI/PstI (*Sec3*), KpnI/EcoRI (*Sec5*), XbaI/BamHI (*Sec6*), KpnI/EcoRI (*Sec8*), and BamHI/XhoI (*Exo70*).

The following hairpin RNA clones were purchased from the Sigma MISSION shRNA library: mouse SEC3 shRNA clones TRCN0000254147, TRCN0000254146, and TRCN0000254145, SEC6 shRNA clones TRCN0000111629, mouse SEC8 shRNA clones TRCN0000307390 and TRCN0000298307, mouse SEC10 shRNA clone TRCN0000093547, and EXO70 shRNA clone TRCN0000376796. Silencing effiiiencies of the shRNA were verified by immunoblotting analysis compared to a control shRNA targeting luciferase. EXO70 sgRNA (5′-CACCGGTTGTCTGGCAGCTGGCTA-3′) was cloned into lentiCRISPR-v2 at the BsmBI restriction site.

For immunoblotting, we used the following antibodies: mouse anti-Sec8 (Clone 14/Sec8, BD Biosciences, Catalog # 610658 1:1000 dilution), mouse anti-Sec6 (Clone 9H5, Novus Biologicals, NBP1-97500, 1:1000 dilution), rabbit anti-Exo70 (Bethyl Laboratories, Catalog # A303-365A, 1:1000 dilution), rabbit anti-GFP (ThermoFisher Scientific, Catalog # A-11122 1:1000 dilution), chicken anti-GFP (Abcam, Catalog # Ab13970, 1:1000 dilution), goat anti-GFP-biotin (Abcam, Catalog # ab6658, 1:2000 dilution), rabbit anti-RFP (Rockland, Catalog # 600–401–379, 1:1000 dilution), mouse anti-α-tubulin (Clone DM1A, Sigma-Aldrich, Catalog # T-9026-1 ML, 1:5000 dilution), rabbit anti-GAPDH (Clone 14C10, Cell Signaling Technology, Catalog # 2118 s, 1:2000 dilution), rabbit anti-Sec3 (Bethyl Laboratories, Catalog # A303–363A, 1:1000 dilution), mouse anti-Sec5 (Clone F-7, Santa Cruz Biotechnologies, Catalog # sc-393230, 1:1000 dilution), mouse anti-Exo84 (Clone H-1, Catalog # 515532, Santa Cruz Biotechnologies, 1:1000 dilution), and GFP-Trap®_MA beads were purchased from ChromoTek GmbH.

Other materials used in this paper were as follows: Q-VD-Oph (Sigma-Aldrich); ProLong® Live Antifade Reagent (ThermoFisher Scientific). Trolox (Acros Organics) was used for live cell imaging. FluoroBright DMEM medium for live cell imaging was from Life Technologies. Fetal bovine serum (FBS) was from Atlantic Biologicals. TetraSpeck Microspheres, 0.02 μm and 0.1 μm (ThermoFisher Scientific). JF585-Halo and JF646-Halo dyes were gifts from Luke Lavis, Janelia Farms HHMI Institute.

**Cell culture and stable cells.** NMuMG (ATCC CRL-1636) and HEK293T (ATCC CRL-3216) cells were obtained from ATCC. Cells were cultured in Dulbecco's modified Eagle medium (Life Technologies), supplemented with 10% FBS and 1X penicillin/streptomycin (Life Technologies) and maintained in culture as suggested by ATCC. NMuMG cells transduced with shRNAs or sgRNA were selected using puromycin (2 μg/ml). Cells infected with VAMP2-pHluorin, VAMP2-Halo, or Akt-Halo were selected using blasticidin (10 μg/ml) or FACS.

**Lentiviral transductions and transient transfections**. Lentivirus was produced by transfecting HEK293T cells with lentiviral packaging vectors pMD2.G and psPAX2 using calcium phosphate precipitation. Lentiviral transductions were performed with virus-conditioned medium collected from HEK293T cells 48 h post transfection. Xfect transfection reagent (Takara) was used to transiently transfect NMuMG cells, to create exocyst knock-in cell lines.

**CRISPR/Cas9-mediated generation of knock-in cell lines**. NMuMG cells were plated at $10^5$ cells/well in a six-well plate 24 h before transfection. Cells were transfected with targeting vector (LITMUS29 backbone), sgRNA (Addgene plasmid #41824), and Cas9 expression vector (pCMVsp6-nls-hCas9-nls), a gift from Li-En Jao (UC Davis), at an equimolar (550 fmol) ratio using Xfect transfection reagent according to the manufacturer's protocol. The GFP-positive cells were subsequently sorted on a FACSAriaIII (BD) 5 d post transfection. After single-cell cloning, insertions were confirmed by immunoblot analysis and PCR-based genotyping. Sequences of sgRNA sequences and primers are available in Supplementary Table 2.

**Immunoprecipitation and quantitative western blot**. NMuMG CRISPR cells lines expressing sfGFP-tagged exocyst subunits were grown to confluency in 100 mm cell culture dishes. Cells were washed with PBS and lysed in 1 ml of lysis buffer containing 20 mM HEPES; pH 7.4, 50 mM NaCl, 2 mM EDTA, and 0.1% Triton X-100, supplemented with cOmplete™ mini EDTA-free protease inhibitor cocktails (Roche) and PhosStop (Roche). Cells were lysed in a rotator for 15 min at 4 °C and debris was removed by centrifugation at 16,000$xg$. For quantitative immunoprecipitation, protein amounts were measured using a BCA assay (ThermoFisher Scientific) and a small amount of lysates from each step (input, unbound, wash) were retained for analysis. Affinity purifications were performed with GFP-Trap_MA beads. Quantitative immunoblot analysis used secondary antibodies conjugated to IRDye® **680**RD or **800**CW (LI-COR) at 1:15,000 dilutions, imaging blots with Odyssey CLx Infrared Scanner (LI-COR). The uncropped immunoblot images are provided in Supplementary Fig 8.

**Affinity purification for mass spectrometry**. Gene-edited NMuMG cell lines were grown to confluency in four to five 150 mm cell culture dishes. Cells were washed three times with 20 ml of PBS and lysed in 10 ml of lysis buffer described above, supplemented with cOmplete™ mini EDTA-free protease inhibitor cocktails (Roche), PhosStop (Roche), and NaF (100 mM; Sigma-Aldrich). Cells were harvested using a cell scraper followed by immediate freezing in liquid $N_2$. Lysates were thawed and briefly sonicated on ice using a microtip sonicator (20 s; Vibra-Cell), followed by centrifugation at 16,000$xg$ for 10 min at 4 °C to remove debris. Supernatants were transferred into 15-ml conical tubes and baits captured on GFP-Trap®_MA beads by end-over-end mixing at 4 °C for 30 min. Beads were separated using Magnetic Particle Concentrator (ThermoFisher Scientific), unbound supernatant was aspirated followed by 3 × 1.5-ml washes in the lysis buffer. Samples were eluted in 2x Laemmli sample buffer and heated to 95 °C for 15 min and subsequently processed for MS analysis.

**Mass spectrometry**. For LC–MS/MS, exocyst protein immunoprecipitations were run ~1.5 cm into a NuPAGE Bis-Tris gel to remove SDS from the samples, were stained with Novex Colloidal Blue Coomassie stain (ThermoFisher Scientific), and destained in water. Coomassie-stained gel regions were cut from the gel and diced into 1-mm³ cubes. Proteins were treated for 30 min with 45 mM DTT, and available Cys residues were carbamidomethylated with 100 mM iodoacetamide for 45 min. Gel pieces were further destained with 50% MeCN in 25 mM ammonium bicarbonate, and proteins were digested with trypsin (10 ng/uL) in 25 mM ammonium bicarbonate overnight at 37 °C. Peptides were extracted by gel dehydration with 60% MeCN, 0.1% TFA, the extracts were dried by speed vac centrifugation, and reconstituted in 0.1% formic acid.

Peptides were analyzed by LC-coupled tandem mass spectrometry (LC-MS/MS). An analytical column was packed with 20 cm of C**18** reverse-phase material (Jupiter, 3μm beads, 300 Å, Phenomenex) directly into a laser-pulled emitter tip. Peptides were loaded on the capillary reverse-phase analytical column (360μm O. D. x 100μm I.D.) using a Dionex Ultimate 3000 nanoLC and autosampler. The mobile-phase solvents consisted of 0.1% formic acid, 99.9% water (solvent A) and 0.1% formic acid, and 99.9% acetonitrile (solvent B). Peptides were gradient-eluted at a flow rate of 350 nL/min, using a 110-min gradient. The gradient consisted of the following: 1–3 min, 2% B (sample loading from an autosampler); 3–88 min, 2–40% B; 88–98 min, 40–95% B; 98–99 min, 95% B; 99–100 min, 95–2% B; 100–110 min (column re-equilibration), 2% B. A Q Exactive Plus mass spectrometer (Thermo Scientific), equipped with a nanoelectrospray ionization source, was used to mass analyze the eluting peptides using a data-dependent method. The instrument method consisted of MS1 using an MS AGC target value of 3e6, followed by up to 20 MS/MS scans of the most abundant ions detected in the preceding MS scan. A maximum MS/MS ion time of 60 ms was used with a MS2 AGC target of 1e5. Dynamic exclusion was set to 15 s, HCD collision energy was set to 28 nce, and peptide match and isotope exclusion were enabled. For

identification of peptides, tandem mass spectra were searched with Sequest (ThermoFisher Scientific) against a *Mus musculus* database created from the UniprotKB protein database (www.uniprot.org). Variable modification of + 15.9949 on Met (oxidation) and + 57.0214 on Cys (carbamidomethylation) were included for database searching. Search results were assembled using Scaffold 4.3.2. (Proteome Software).

For MRM–MS, peptides for each protein were selected based on their appearance in data-dependent analyses and then optimized for the most useful transitions to monitor. Heavy-labeled peptide internal standards were synthesized by jpt (SpikeTides TQL peptides, jpt, Berlin, Germany) which contained isotopically labeled terminal arginine or lysine residues ($^{13}$C and $^{15}$N) and a trypsin-removable C-terminal tag (Supplementary Table 2). These isotopically labeled peptides were digested separately and then spiked into samples at approximately endogenous levels after in-gel digestion of sample proteins. Skyline software (University of Washington, MacCoss lab) was used to set up scheduled, targeted MRM methods monitoring four to five MRM transitions per peptide. A final MRM instrument method including the isotopically labeled standards and encompassing a 9-min window around the retention time of each peptides was performed using a 40 mm by 0.1 mm (Jupiter 5 micron, 300 A) kasil fritted trap followed by a 250 mm by 0.1 mm (Jupiter 3 micron, 300 A), self-packed analytical column coupled directly to a TSQ-Vantage (ThermoFisher) via a nanoelectrospray source. Peptides were resolved using an aqueous to organic gradient flowing at 400 nl min⁻¹. Q1 peak width resolution was set to 0.7, collision gas pressure was 1 mTorr, and utilized an EZmethod cycle time of 3 s.

**Single-molecule experiments**. No. 1.5 glass coverslips (Electron Microscopy Sciences) and quartz slides (Ted Pella Inc.) were rinsed in 100% ethanol, followed by acetone and sonicated for 15 min. Coverslips and slides were then put in 1 M KOH (Sigma-Aldrich) and sonicated for an additional 30 min, followed with three rinsed in ultrapure water, and additional rinses in acetone. The materials were further cleaned in 0.5 vol% Helmanex III solution (Sigma-Aldrich) for 30 min followed by rinsing in ultrapure water three times. Surfaces were then sonicated for 10 min in 99.9% HPLC-grade methanol (Fisher Chemical) and surfaces torched by swiping three times over a Bunsen burner. Finally, any remaining residual impurities were removed by cleaning the surfaces using a plasma cleaner (PDC-32G, Harrick Plasma).

Approximately 300,000 cells from 24-well plates were used. Cells were first labeled with JF**585**-Halo dye for 90 min, followed by extensive 5x washes with PBS to remove any unbound dye. Cells were then lysed in ice-cold lysis buffer (20 mM Hepes, 50 mM NaCl, 2 mM EDTA, and 0.15% Triton X-100), centrifuged at 13,000$xg$ for 5 min to remove debris, and a fraction of the cleared lysate was spread in between clean quartz slides and a glass coverslip for imaging. Images were taken using a Nikon TIRF microscope, Apo-TIRF 100 × /1.49 oil immersion lens online with a Photometrics Prime 95B sCMOS camera. Both GFP and JF585-Halo were simultaneously excited using AOTF- driven Nikon LUN-F multi-excitation diode laser lines firing 488-and 561-nm lasers and images collected on the camera using an Optosplit III setup (Cairn Research). Images were taken at 12.5 Hz for 40 s. Photobleaching steps were analyzed in MATLAB[49]. Fraction bound was determined according to colocalization of green and red particles, analyzed in FIJI.

To determine the efficiency of HaloTag labeling in NMuMG cells, we created a Venus-HaloTag fusion construct in lentiviral vector pLV-Venus. HaloTag coding sequence was inserted C-terminally to Venus between restriction sites BamHI/NdeI. Venus-Halo was expressed in NMuMG cells to test HaloTag labeling efficiency on incubation for 1.5 h with different concentrationS of JF585-labeled HaloTag ligand, or incubating cells with a single concentration of the ligands for 1 or 18 h. Labeling efficiency ($L_{eff}$) was determined by spreading the cleared cell lysates as described above and calculated according to the expression:

$$\mathbb{L}_{eff} = \frac{1}{n}\sum_{i=1}^{n}\left(\frac{JFdye_{Halo}}{JFdye_{Halo} \cup Venus}\right)_i \qquad (1)$$

To determine the point spread function (PSF) of the TIRF objective with numerical aperture (NA) of 1.49, we used 20- and 100-nm TetraSpeck beads (ThermoFisher Scientific). Beads were spread on a glass coverslip and allowed to settle before images were taken and analyzed using the Born & Wolf PSF model as described elsewhere[50], and using an open-source MATLAB package (http://bigwww.epfl.ch/algorithms/psfgenerator/).

**Live cell TIRFM imaging and analysis**. TIRF images were collected with an Andor Zyla sCMOS on Nikon Ti-E or a Photometrics Prime 95B mounted on a Nikon Ti-2 microscope, using an Apo TIRF 60 × /1.49 oil immersion lens. The Nikon Ti-E TIRF microscope was equipped with multi-excitation diode laser lines (405 nm, 488 nm, 561 nm, and 647 nm) with triggering capabilities for fast dual-color acquisitions, as described above. Image acquisitions were performed using NIS Elements Advanced Research imaging software [version 4.60.00 (Build 1171) Patch 02]. Images were processed and analyzed with Elements or FIJI.

For analysis of single-fusion events, each acquired image sequence was manually reviewed multiple times to visually identify vesicle fusion events as demarcated by Vamp2-pHluorin or TfR-pHuji fluorescence intensity flashes. Coordinates of the vesicle fusion events were marked by a circular ROI. GFP or mApple particle durations were analyzed using NIS-Elements. Circular ROIs ~1μm were drawn inside the initial ROIs and were used to calculate the intensities of sfGFP and pHuji. Arrival and departure of sfGFP were determined based on their increase and decrease in intensities, respectively. Kymographs of sfGFP and pHuji were generated and analyzed using NIS-Elements.

Particle coincidences were measured using NIS-Element's Binary/spot detection algorithm. Spot diameter parameters were set between 0.27 and 0.35 μm, and the contrast was adjusted to detect objects above background for these analysis. The number of spots that were coincident was measurement subsequently.

Photobleaching step detection in Fig. 4 and Fig. S4 was done in MATLAB[49]. Codes are available on Github and access information provided in the Data Availability section. Exocyst molecule numbers were estimated from GFP intensity standards. One, two, or three sfGFP were fused to the C-terminus of monomeric CD86 and expressed in HEK293T cells[51]. Intensities were measured at a fixed optimal camera and laser setting. Intensities of the molecules were matched to photobleaching steps and fitted to a linear regression model. The same instruments settings and the region of the image with uniform intensities were then used to measure exocyst molecule intensities during vesicle fusions. The peak intensities of single-resolved sfGFP-exocyst particles at vesicle fusion sites, indicated by TfR-pHuji, were then measured and the number of molecules/particle was determined from the regression equation. Supplementary Fig. 7e–i further explains the criterions used for fluorescence intensity measurements.

Mean squared displacements of the different exocyst subunits from TIRFM images were measured using Imaris image analysis software (Bitplane) tracking algorithm. For two-color tracking, each channel was individually processed and overlaid to assess differences in overall diffusion patterns.

**Dual-color fluorescence cross-correlation spectroscopy in live cells**. dcFCS setup and measurements were performed using a custom-built confocal microscope with multi-color excitation and detection and real-time multi-channel correlation analysis capabilities. sfGFP and JF646 were excited simultaneously by a blue laser (TECBL—488 nm, WorldStarTech) and a red laser (TECRL—633 nm, WorldStarTech). Excitation power at the sample was set in the range of 0.1–5μW, corresponding to an average intensity of 0.05–2.5 kW$^{-1}$cm$^2$ in the diffraction-limited confocal area. Fluorescence signals from sfGFP and JF646 were separated by a dichroic filter (FF585-Di01, Semrock) and directed through a green band-pass filter (FF01–512/25, Semrock) in the case of sfGFP or a red long-pass filter (LP02-647RS, Semrock) in the case of JF646. Each signal was split using a 50/50 non-polarizing beamsplitter. The four signals were collected using four avalanche photodiode (APD) detectors (SPCM-AQR-13-FC, PerkinElmer Inc.) and processed using a multichannel hardware correlator (Flex11–8Ch, correlator.com). The correlator outputs up to four different intensity correlation curves between any two detection channels. For typical live cell measurements, at least 10$^7$ photons were collected from each cell to construct the FCS curves.

Live cell measurements—Experiments on NMuMG cells were performed using glass-bottom Petri dishes (MatTek). All imaging was performed in FluoroBrite DMEM media (ThermoFisher Scientific) supplemented with Trolox (2 mM). Confocal scan imaging was employed to search for cells that exhibit an optimal fluorescence signal of 5–100 kHz at a laser excitation intensity of 50–250 W$^{-1}$cm$^2$.

**Model fitting for FCS**. Correlation curves from dcFCS were analyzed in MATLAB using a custom-written program based on the Marquardt–Levenburg algorithm[52,53]. The data were weighted according to the standard deviation at each point, as determined from multiple repeated readings. Rhodamine 110 and Alexa Fluor 647 were measured in an aqueous PBS buffer to calibrate geometrical parameters (lateral radius, $w_0$, and aspect ratio, $s$) of the confocal detection volumes, which were approximated as 3D Gaussians.

For exocyst proteins diffusing in live cells, a two-component diffusion model was applied to fit the three correlation curves, namely green autocorrelation (for sfGFP), red autocorrelation (for JF-646), and cross-correlation:

$$G_j(\tau) = G_j(0)\Big(f_{D1} \cdot Diff_{D1j}(\tau) + f_{D2} \cdot Diff_{D2}(\tau)\Big), j = \text{green or red} \tag{2}$$
$$G_x(\tau) = G_x(0) \cdot Diff_{D2}(\tau)$$

where the diffusion terms are expressed as

$$Diff_{D1j} = \left(1 + \frac{\tau}{\tau_{D1j}}\right)^{-1}\left(1 + \frac{1}{s^2} \times \frac{\tau}{\tau_{D1j}}\right)^{-1/2}, j = \text{green or red} \tag{3}$$
$$Diff_{D2} = \left(1 + \frac{\tau}{\tau_{D2}}\right)^{-1}\left(1 + \frac{1}{s^2} \times \frac{\tau}{\tau_{D2}}\right)^{-1/2}$$

for three-dimensional diffusion in the cytosol, or

$$Diff_{D1j} = \left(1 + \frac{\tau}{\tau_{D1j}}\right)^{-1}, j = \text{green or red}$$
$$Diff_{D2} = \left(1 + \frac{\tau}{\tau_{D2}}\right)^{-1} \tag{4}$$

for two-dimensional diffusion at the membrane.

Note that the cross-correlation contains only one diffusion component, which has the same parameters as the slow diffusion component of the green and the red autocorrelation curves. In Eqs. 2–4, $\tau_{D1}$ and $\tau_{D2}$ are the residence times of the two diffusing species, and $f_{D1}$ and $f_{D2}$ are corresponding fluorescence intensity contributions of the two diffusion species ($f_{D1} + f_{D2} = 1$). In the case of singly labeled proteins, which we measure here, $f_{D1}$ and $f_{D2}$ equal to the concentration fractions (mole fractions) of the two diffusing species. Diffusion coefficients ($D$) were calculated from fitted estimate of $\tau_D$ values as

$$D = w_0^2/4\tau_D \tag{5}$$

where $w_0$ is the lateral radius of the detection volume. Fractions of coupling are then calculated from the correlation amplitudes after fitting to Eq 2:[54]

$$\text{fraction of green} = \frac{G_x(0)}{G_r(0)}$$
$$\text{fraction of red} = \frac{G_x(0)}{G_g(0)} \tag{6}$$

Fractions of coupling calculated from Eq. 6 were corrected for spectral cross-talk using available fluorescence intensity information[55]. Correction factors were consistently lower than 0.1% owing to the high fluorescence intensity of the JF-646 dye relative to GFP. In addition, the fractions were not corrected for possible detection volume misoverlap due to the lack of a reliable in-cell control sample. This may lead to an underestimation by a constant scaling factor, which does not affect the comparison of measured fractions of coupling across different cells/samples.

The hydrodynamic radius ($R_H$) of the fluorescent particles measured by FCS was estimated from the diffusion coefficient using the Stokes–Einstein equation:

$$R_H = \frac{k_B T}{6\pi\eta D} \tag{7}$$

where $k_B$ is the Boltzmann constant, $T$ is the measurement temperature (20 °C), $\eta$ is the viscosity of the solvent (viscosity of water was used for the calculation), and $D$ is the diffusion coefficient estimated from the autocorrelation curve.

**Statistical analysis**. Data are reported as ±SD, ±SEM, or 25th−75th IQR, and analyzed by Student's $t$ test or one-way analysis of variance (ANOVA) as indicated using Graphpad Prism 7, RStudio, or JASP (ver.0.9, University of Amsterdam) statistical softwares. For curve fitting, fits were assessed by accounting for variation in the FCS data curve and chi-square test statistics. When using one-way ANOVA or Kruskal–Wallis test, post hoc analysis was done using Tukey's, Dunn's, or Scheffe's multiple comparison tests as appropriate. All statistical analysis was considered significant at $p < 0.05$. Effect size in terms of Cohen's $d$ was calculated using JASP.

## Data availability

All data generated or analyzed during this study are included in this published article (and its Supplementary Information Files), or from the authors upon reasonable request. The mass spectrometry proteomics data have been deposited to the ProteomeXchange Consortium (http://proteomecentral.proteomexchange.org) via the PRIDE partner repository with the dataset identifier PXD011370 [http://proteomecentral.proteomexchange.org/cgi/GetDataset?ID = PXD011370]. MATLAB scripts used in the paper to analyze dcFCS and single-molecule photobleaching data are available in a public repository at https://github.com/mukhtar06/singlemolecule.

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

## Acknowledgements

This work was supported by NIH/NIGMS grant (GM070902) to I.G.M., NSERC Discovery Grant (RGPIN 2017–06030) to C.C.G., postdoctoral fellowships from the CIHR to S.M.A., and Grant-in-Aid for JSPS Research Fellow (17J40028) to H.N.F. Experiments and data analysis were performed in part through the use of the Vanderbilt Cell Imaging Shared Resources. Mass spectrometry experiments were performed using Vanderbilt MSRC Proteomics Lab. We would like to thank Jorge Rua Fernandez, Rabindra Shivnaraine, Goker Arpag, Marija Zanic, Tomas Kirchhausen, and members of the Macara lab for helpful insights. We also thank Yasutsugu Takada, Shigeki Higashiyama, and Shinji Fukuda (Ehime University, Japan) for their valuable support.

## Author contributions

S.M.A., H.N.F., Y.L., and W.H.M. conducted and analyzed the experiments. S.M.A., H.N.F., and I.G.M. designed the experiments. Y.L and C.G performed FCCS data analysis. S.M.A and I.G.M. wrote the paper.

## Additional information

**Competing interests:** The authors declare no competing interests.

