## [Peer Review File · Nature Communications]

Reviewers' comments:

Reviewer #1 (Remarks to the Author):

This paper exploits endogenously expressed fluorescent tags and advanced light microscopy techniques to describe the assembly dynamics of the exocyst vesicle tethering complex in mammalian cells. In total the studies provide support for a model that is actually very similar to the current model of yeast exocyst complex assembly. There are certainly some differences and refinements, but none that fundamentally change our concepts about what the exocyst is doing to facilitate vesicular traffic. Perhaps the most novel finding is that the two exocyst subcomplexes can dissociate to some extent in the cytoplasm and can be independently recruited to secretory vesicles where they assemble prior to fusion. Nonetheless it is not clear what the implications are regarding function. It has not been established that there is an obligatory assembly/disassembly cycle that is tightly coupled to rounds of vesicle traffic rather than a steady state of association and dissociation in which the affinity increases to some extent as the complex interacts with additional components on the vesicle surface. The work is of a high caliber and the authors have made the most of the tools available, but without explicitly establishing the relevance of assembly dynamics to function the study remains largely descriptive in nature and may be better suited for a more specialized journal.

Specific comments:

1. In Figure 1C it appears that the tagged alleles of Exo70 and Sec8 are overexpressed relative to the endogenous copies. Have multiple integrants been tested and is this overexpression consistently observed? Overexpression, even by a couple of fold, will interfere with the interpretation of assembly studies.
2. A coomassie stained SDS gel of the pulldowns would be very helpful to support the MS data in Fig 1E.
3. In Fig 2A it might be informative to include data concerning the various exocyst interacting proteins from Fig 1E.
4. Fig 2C, it would help to also show an image from the cell center for comparison.
5. In their analysis of Fig 2 the authors distinguish between vesicle association and PM association. It is not stated how this distinction is made. This is critical for some of their conclusions.
6. In Fig 3 two different constructs are used to reveal fusion events, one uses a fusion to Vamp2 the other to TfR. Why are different constructs used, do they mark the same vesicles and do they give the same results?
7. While the statistical analysis is generally very thorough, I was not able to ascertain if Sec3 always leaves the fusion site before the other subunits or just on average leaves before the other subunits. Similarly is there a definitive order regarding the time of vesicle association of the two subcomplexes or just an average difference?
8. In Fig 5 E it might be informative to extend the analysis to the other members of the two subcomplexes.

Reviewer #2 (Remarks to the Author):

The paper submitted by Ahmed and Nishida-Fukuda et al. is a step forward for the understanding of a mechanism of vesicle tethering and should be published in this journal. Here the authors study the dynamics of vesicle tethering on a time scale fast enough to observe differences between the subunits of exocyst. With the recent technologies of sCMOS cameras and CRISPR/Cas9 gene editing coupled with vesicle fusion markers the authors were able to do what other groups have been unable, see fusion from the perspective of each exocyst subunit. The high frame rate reveals that Sec3 of exocyst in fact departs the membrane before the other subunits. Why Sec3 makes an early departure remains a mystery.

Here

the authors were able track subunits of the exocyst complex allowing them to answer if subunits arrive separately or together at the plasma membrane, as well as tracking Rab11 on vesicles. They conclude from the four subunits of exocyst tested that exocyst arrives at the plasma membrane with the vesicle rather than being assembled on the plasma membrane. Additionally the authors find that if we want to be understanding more about the mysterious properties of exocyst, we should be looking to Sec3. They show not only does it leave the plasma membrane before fusion, it also is not needed to maintain the Sec5/6/8 heterotrimer (part of SC1) once it has formed. And in contrast to the Picco paper from 2017, the authors find that about 9 exocyst complexes associate with a fusing vesicle, and perhaps the conflicting numbers are from this being the mammalian system instead of budding yeast.

One

conclusion, that half of the exocyst subunits are assembled in the cytosol while the other subunits are not in a stable complex, leaving less than 50% of octomer complex in the cytosol, adds more confusion than answers. First, what percentage of intact protein complex constitutes a "stable" complex? Second, as was mentioned in the manuscript the Cryo EM structure was recently solved, as was a negative stain structure a few years before, both showing great similarity. Careful reading of those papers would reveal that the purification steps involved clearing the membrane either without or with low concentrations of detergents, and the remaining lysates were then used for affinity tag based purification and structure determination. Yet, the structures reveal a eight component complex, this implies that (at least in the yeast systems used in these papers) that exocyst remains an intact eight protein complex in the cytosol, differing from the results presented here. A recently submitted Biorxiv paper shows negative stain class averages of the mammalian exocyst complex show that it is at least in the global structure very similar to the published structures from yeast, implying that mammalian exocyst may also have a stable state. The authors should address these points in the text, specifically how their results are or are not consistent with the published structures.

I

believe that part of this confusion between in vivo and in vitro systems will be settled once we can understand what an "active complex" looks like. The structures solved were bound to no partners which are needed in the steps of vesicle trafficking and fusion, and these partners could greatly affect the composition of the exocyst complex. Most likely the structures that have been solved are of an "inactive complex" in which all eight exocyst proteins are together but are not yet primed for vesicle trafficking and fusion.

The

authors offer that in future studies they hope to understand the effect of SNARE as well as the many other interacting proteins, with the exocyst. I greatly look forward to seeing these papers. The more we can know about effectors of exocyst the more we can understand how this protein complex works to carry out the all important task of vesicle fusion.

Reviewer #3 (Remarks to the Author):

The article by Ahmed et al. addresses the question of the assembly of the exocyst at vesicles and the plasma membrane in vesicle fusion. For that purpose, the authors use a range of single molecule fluorescence techniques in live cell experiments. They demonstrate that two tetrameric subcomplexes assemble to build the exocyst and that these subcomplexes can independently

associate with membranes but are both needed for fusion. They find, interestingly, different dynamical behavior of the subcomplexes, with SC1 and SC2 arriving at slightly different times at the plasma membrane, indicating that assembly happens at the plasma membrane. For Sec3, part of SC1, they find a more dynamical behavior indicating a lower affinity. Their analysis leads to a refined model of the exocyst assembly and function in mammalian cells. The article is clearly written and the data supports their conclusions. However, some points require clarification or correction.

Page 10 and 11: The authors interpret 40% cross-correlation amplitude for YFP-Halo as being equivalent to the 40% fractional labeling efficiency. However, that is only true if the maximum cross-correlation achievable in the system is known. For that, a positive control for FCCS is required. In the same vein, the authors see about 70% cross-correlation for Sec8-GFP/Sec5-Halo. To evaluate what this corresponds to in actual interactions, the authors would need a positive control to determine what the maximum cross-correlation is (see point 1). The cross-correlation amplitude is limited by a multitude of parameters, including the overlap of the emission volumes of the two dyes, possible chromatic aberrations and shifts of the volumes, dark states and maturation/folding issues of the fluorescent proteins, and/or FRET. The authors comment on the non-correction for volume overlap on page 34. While that does not change comparison of relative fractions, it still means that absolute fractions cannot be determined. Thus, the 40% cross-correlation for Sec8-GFP/Sec5-Halo supports suboptimal labeling but cannot corroborate the absolute values. And the 70% cross-correlation for Sec8-GFP/Sec5-Halo, is likely an underestimate and could correspond to any absolute interaction from 70-100%.

Page 12: The authors don't state the determined PSF size, nor the relevant size of the pixels. So one cannot evaluate whether the 4 pixel distance is justified (even if one tries to estimate the values from the given objective and camera). It would be easier for the reader if the data is provided.

Page 34, equation 1: The equation is correct if the brightness of both diffusing species is the same. If the brightness is not the same, the fluorophores contribute with the square of their brightness. Therefore, the statement on page 35 is not correct that $fD1 + fD2 = 1$ if $fD1$ and $fD2$ are fluorescence contributions. $fD1 + fD2 = 1$ holds only if $fD1$ and $fD2$ represent mole fractions. This, however, does not change the interpretation of the data as the authors see singly labeled species the brightness for both particles is the same.

Minor issues:

Page 9: "Nevertheless, once at the PM SC and 2 remain ..." is missing a comma and should better read "Nevertheless, once at the PM, SC and 2 remain ..."

Page 12: PFS should read PSF (was that defined anywhere?)

Page 14: "perhaps from subcomplexes" should read "perhaps from subcomplexes"

Reviewer comment

Reviewer #1 (Remarks to the Author):

This paper exploits endogenously expressed fluorescent tags and advanced light microscopy techniques to describe the assembly dynamics of the exocyst vesicle tethering complex in mammalian cells. In total the studies provide support for a model that is actually very similar to the current model of yeast exocyst complex assembly. There are certainly some differences and refinements, but none that fundamentally change our concepts about what the exocyst is doing to facilitate vesicular traffic. Perhaps the most novel finding is that the two exocyst subcomplexes can dissociate to some extent in the cytoplasm and can be independently recruited to secretory vesicles where they assemble prior to fusion. Nonetheless it is not clear what the implications are regarding function. It has not been established that there is an obligatory assembly/disassembly cycle that is tightly coupled to rounds of vesicle traffic rather than a steady state of association and dissociation in which the affinity increases to some extent as the complex interacts with additional components on the vesicle surface. The work is of a high caliber and the authors have made the most of the tools available, but without explicitly establishing the relevance of assembly dynamics to function the study remains largely descriptive in nature and may be better suited for a more specialized journal.

Specific comments:

1. In Figure 1C it appears that the tagged alleles of Exo70 and Sec8 are overexpressed relative to the endogenous copies. Have multiple integrants been tested and is this overexpression consistently observed? Overexpression, even by a couple of fold, will interfere with the interpretation of assembly studies.
2. A coomassie stained SDS gel of the pulldowns would be very helpful to support the MS data in Fig 1E.
3. In Fig 2A it might be informative to include data concerning the various exocyst interacting proteins from Fig 1E.
4. Fig 2C, it would help to also show an image from the cell center for comparison.
5. In their analysis of Fig 2 the authors distinguish between vesicle association and PM association. It is not stated how this distinction is made. This is critical for some of their conclusions.
6. In Fig 3 two different constructs are used to reveal fusion events, one uses a fusion to Vamp2 the other to TfR. Why are different constructs used, do they mark the same vesicles and do they give the same results?
7. While the statistical analysis is generally very thorough, I was not able to ascertain if Sec3 always leaves the fusion site before the other subunits or just on average leaves before the other subunits. Similarly is there a definitive order regarding the time of vesicle association of the two subcomplexes or just an average difference?
8. In Fig 5 E it might be informative to extend the analysis to the other members of the two subcomplexes.

Point by point response to reviewer comments

Authors' response to Reviewer 1

This reviewer appreciated the quality of our study, but feels that overall the conclusions support a model similar to the current model for yeast exocyst assembly, and do not fundamentally change our concepts about what the exocyst is doing to facilitate vesicular traffic.

To a point we have to agree – clearly, we have not revolutionized an understanding of how the exocyst works. However, as we explain in the introduction, there have been multiple, mutually incompatible models for how and where the yeast exocyst complex assembles and disassembles, which have not yet been fully reconciled; and the data for the mammalian complex is extremely sparse. Therefore, we argue that our study provides important new information on exocyst biology, and also sets a standard for future studies (not only of the exocyst but for other multisubunit complexes).

Specifically, we demonstrate that there are two exocyst subcomplexes that exist independently as well as in the octameric complex (which to our knowledge has not yet been previously shown in yeast); and that these subcomplexes can associate separately with the plasma membrane (which has also not been shown in yeast). We also provided the novel finding that Sec3 preferentially disassembles from the complex prior to fusion; and now add data suggesting a bias in the arrival at the PM of SC2 prior to SC1. Together, we believe that these observations provide important new information on the dynamics and function of the exocyst.

Specific points:

We thank the reviewer for raising a number of important issues that we have addressed as outlined below. We believe that the changes we have made in response to these comments have significantly strengthened the manuscript.

1. In Figure 1C it appears that the tagged alleles of Exo70 and Sec8 are overexpressed relative to the endogenous copies. Have multiple integrants been tested and is this overexpression consistently observed? Overexpression, even by a couple of fold, will interfere with the interpretation of assembly studies.

Please note that the figure was not meant to be quantitative, but rather to confirm that the tagged proteins were of the expected size and the clones were homo- or heterozygous. Therefore, loading between lanes was not carefully quantified. However, normalizing to loading controls shows no difference between tagged/untagged Exo70 or Sec8. To make this clear we have repeated the immunoblots for Exo70 and Sec8 using equal loading controls, and now use these data in Figure 1c. There were no significant differences between different integrants (not shown).

2. A Coomassie stained SDS gel of the pulldowns would be very helpful to support the MS data in Fig 1E.

Although we ran the precipitated samples on a gel, this was performed to remove Laemmli sample buffer, not for protein separation, and was stopped when the dye front had moved only 1.5 cm into the separating gel. The entire region of the lane containing the sample was excised, diced and trypsinized to recover peptides for mass spectrometry runs. We have now clarified this in the methods section (page 17).

3. In Fig 2A it might be informative to include data concerning the various exocyst interacting proteins from Fig 1E.

It would be very interesting in a future study to know which accessory proteins associate with each subcomplex. However, the major difference between the experiment in Fig 1E and Fig 2A is that the former experiment was done for protein (peptide) discovery, whereas the latter was to quantify the abundance of the peptides using MRM-MS and full-scan MS. The stoichiometry of interaction for the other interacting partners and their absolute amounts shown in Fig 1E and

Supplementary Table 1, were not generally high enough to provide quantitative measures. Since this experiment was initially designed to address the interactions between the exocyst subunits, we have limited our quantitative threshold to that of the exocyst.

4. Fig 2C, it would help to also show an image from the cell center for comparison.

A valid criticism is that we do not specifically show that the cell imaged in Figure 2C actually expresses Exo70-sfGFP – mainly because of technical issues, since our TIRF system is not attached to a confocal microscope so we cannot image the center of the same cell. We are using clonal cell lines, and 100% of the cells express tagged Exo70. Nonetheless, it is interesting to ask if the tagged protein is destabilized and degraded when a subunit from the same subcomplex is depleted. Although we do not have epifluorescence images to address this point we have confirmed by immunoblot that the expression levels of other proteins in a subcomplex are not depleted when one subunit is silenced (see Supplementary Fig 2i; and new text, page 6).

5. In their analysis of Fig 2 the authors distinguish between vesicle association and PM association. It is not stated how this distinction is made. This is critical for some of their conclusions.

Observation of any particle in the TIRF field (~100 nm) was considered to be membrane-associated. To distinguish exocyst attached to vesicles versus associated just with the plasma membrane, we measured the coincidence between exocyst-GFP tagged particles and mApple-Rab11 labeled vesicles, which in control cells was ~70% (shown in Supplementary Fig 2l-n).

6. In Fig 3 two different constructs are used to reveal fusion events, one uses a fusion to Vamp2 the other to TfR. Why are different constructs used, do they mark the same vesicles and do they give the same results?

We find that in general, pHuji is not as robust a pH sensor as pHluorin. In our hands TfR-pHuji consistently produced flashes during vesicle fusions, but this was not the case using Vamp2-pHuji, perhaps because it is expressed at a lower level. Nevertheless, our tests showed that TfR-pHuji positive vesicles are also Vamp2-pHuorin positive, an example of which is now shown in a new figure panel Fig 3d.

7. While the statistical analysis is generally very thorough, I was not able to ascertain if Sec3 always leaves the fusion site before the other subunits or just on average leaves before the other subunits. Similarly is there a definitive order regarding the time of vesicle association of the two subcomplexes or just an average difference?

We refer the reviewer to the graph in Fig 3h. In about half of our observations, Sec3 left the vesicle docking site prior to or at the same time as fusion. The other 50% left before fusion.

The second part of the question is interesting and we apologize for omitting our data concerning this point in the original manuscript. While single molecule/single particle interactions are rarely all-or-none, we did see a clear bias towards the prior arrival of SC2 (54% versus 7% for SC1). About 40% of arrivals were coincident within the time resolution of our camera. These data are now shown in a new Fig 4c, and are discussed in the text.

8. In Fig 5 E it might be informative to extend the analysis to the other members of the two subcomplexes.

We agree with the reviewer that extending the analysis to the other members of the two subcomplexes would be informative, and we attempted to create the necessary double-tagged cell lines. Unfortunately, despite several attempts, we were not able to successfully isolate clones for these other subunit combinations, presumably because the exocyst complex cannot assemble or is non-functional.

Reviewer #2 (Remarks to the Author):

The paper submitted by Ahmed and Nishida-Fukuda et al. is a step forward for the understanding of a mechanism of vesicle tethering and should be published in this journal. Here the authors study the dynamics of vesicle tethering on a time scale fast enough to observe differences between the subunits of exocyst. With the recent technologies of sCMOS cameras and CRISPR/Cas9 gene editing coupled with vesicle fusion markers the authors were able to do what other groups have been unable, see fusion from the perspective of each exocyst subunit. The high frame rate reveals that Sec3 of exocyst in fact departs the membrane before the other subunits. Why Sec3 makes an early departure remains a mystery.

Here the authors were able track subunits of the exocyst complex allowing them to answer if subunits arrive separately or together at the plasma membrane, as well as tracking Rab11 on vesicles. They conclude from the four subunits of exocyst tested that exocyst arrives at the plasma membrane with the vesicle rather than being assembled on the plasma membrane. Additionally the authors find that if we want to be understanding more about the mysterious properties of exocyst, we should be looking to Sec3. They show not only does it leave the plasma membrane before fusion, it also is not needed to maintain the Sec5/6/8 heterotrimer (part of SC1) once it has formed. And in contrast to the Picco paper from 2017, the authors find that about 9 exocyst complexes associate with a fusing vesicle, and perhaps the conflicting numbers are from this being the mammalian system instead of budding yeast.

One conclusion, that half of the exocyst subunits are assembled in the cytosol while the other subunits are not in a stable complex, leaving less than 50% of octomer complex in the cytosol, adds more confusion than answers. First, what percentage of intact protein complex constitutes a “stable” complex? Second, as was mentioned in the manuscript the Cryo EM structure was recently solved, as was a negative stain structure a few years before, both showing great similarity. Careful reading of those papers would reveal that the purification steps involved clearing the membrane either without or with low concentrations of detergents, and the remaining lysates were then used for affinity tag based purification and structure determination. Yet, the structures reveal a eight component complex, this implies that (at least in the yeast systems used in these papers) that exocyst remains an intact eight protein complex in the cytosol, differing from the results presented here. A recently submitted Biorxiv paper shows negative stain class averages of the mammalian exocyst complex show that it is at least in the global structure very similar to the published structures from yeast, implying that mammalian exocyst may also have a stable state. The authors should address these points in the text, specifically how their results are or are not consistent with the published structures.

I believe that part of this confusion between in vivo and in vitro systems will be settled once we can understand what an “active complex” looks like. The structures solved were bound to no partners which are needed in the steps of vesicle trafficking and fusion, and these partners could greatly affect the composition of the exocyst complex. Most likely the structures that have been solved are of an “inactive complex” in which all eight exocyst proteins are together but are not yet primed for vesicle trafficking and fusion.

The authors offer that in future studies they hope to understand the effect of SNARE as well as the many other interacting proteins, with the exocyst. I greatly look forward to seeing these papers. The more we can know about effectors of exocyst the more we can

understand how this protein complex works to carry out the all important task of vesicle fusion.

Authors' responses to Reviewer 2

We greatly appreciate the positive comments of this reviewer. We have modified the Discussion to address where our data are consistent or not with the known exocyst structures. Concerning the percentage of protein in a 'stable' complex, we assume that the octamer is in a dynamic equilibrium with free subcomplexes. Please note, however, that based on a comment from Reviewer #3 we have corrected the FCCS data for the maximum detectable cross-correlation, derived from measurements on the YFP-Halo fusion protein. This correction increases the measured abundance of octamer in the cell to about 65%, and reduces subcomplexes and free subunit abundance to ~25% and ~10% respectively (see modified Figures 6 and 8). These corrections do not change our conclusions but do support the idea that within the cell the dynamic equilibrium between subunits favors the full octameric complex.

We have also added new information (Fig. 4c) showing a bias towards the first arrival of SC2, followed by (or simultaneous with) SC1, suggesting that there is a preferred order of assembly of a functional exocyst at the PM.

Reviewer #3 (Remarks to the Author):

The article by Ahmed et al. addresses the question of the assembly of the exocyst at vesicles and the plasma membrane in vesicle fusion. For that purpose, the authors use a range of single molecule fluorescence techniques in live cell experiments. They demonstrate that two tetrameric subcomplexes assemble to build the exocyst and that these subcomplexes can independently associate with membranes but are both needed for fusion. They find, interestingly, different dynamical behavior of the subcomplexes, with SC1 and SC2 arriving at slightly different times at the plasma membrane, indicating that assembly happens at the plasma membrane. For Sec3, part of SC1, they find a more dynamical behavior indicating a lower affinity. Their analysis leads to a refined model of the exocyst assembly and function in mammalian cells. The article is clearly written and the data supports their conclusions. However, some points require clarification or correction.

Page 10 and 11: The authors interpret 40% cross-correlation amplitude for YFP-Halo as being equivalent to the 40% fractional labeling efficiency. However, that is only true if the maximum cross-correlation achievable in the system is known. For that, a positive control for FCCS is required. In the same vein, the authors see about 70% cross-correlation for Sec8-GFP/Sec5-Halo. To evaluate what this corresponds to in actual interactions, the authors would need a positive control to determine what the maximum cross-correlation is (see point 1). The cross-correlation amplitude is limited by a multitude of parameters, including the overlap of the emission volumes of the two dyes, possible chromatic aberrations and shifts of the volumes, dark states and maturation/folding issues of the fluorescent proteins, and/or FRET. The authors comment on the non-correction for volume overlap on page 34. While that does not change comparison of relative fractions, it still means that absolute fractions cannot be determined. Thus, the 40% cross-correlation for Sec8-GFP/Sec5-Halo supports suboptimal labeling but cannot corroborate the absolute values. And the 70% cross-correlation for Sec8-GFP/Sec5-Halo, is likely and underestimate and could correspond to any absolute interaction from 70-100%.

Page 12: The authors don't state the determined PSF size, nor the relevant size of the pixels. So one cannot evaluate whether the 4 pixel distance is justified (even if one tries to estimate the values from the given objective and camera). It would be easier for the reader if the data is provided.

Page 34, equation 1: The equation is correct if the brightness of both diffusing species is the same. If the brightness is not the same, the fluorophores contribute with the square of their brightness. Therefore, the statement on page 35 is not correct that $fD1+fD2=1$ if $fD1$ and $fD2$ are fluorescence contributions. $fD1+fD2=1$ holds only if $fD1$ and $fD2$ represent mole fractions. This, however, does not change the interpretation of the data as the authors see singly labeled species the brightness for both particles is the same.

Minor issues:

Page 9: "Nevertheless, once at the PM SC and 2 remain ..." is missing a comma and should better read "Nevertheless, once at the PM, SC and 2 remain ..."

Page 12: PFS should read PSF (was that defined anywhere?)

Page 14: "perhaps from subcomplexes" should read "perhaps from subcomplexes"

Point by point response to reviewer comments

Authors' responses to Reviewer 3.

We thank this reviewer for their careful review of the FCCS data and in response we have introduced a correction for the maximum coupling determined from the covalently linked YFP-Halo fusion protein. This correction increases the estimated abundance of the octameric complex in the cell, but does not change any of the conclusions of the study.

1. Page 10 and 11: The authors interpret 40% cross-correlation amplitude for YFP-Halo as being equivalent to the 40% fractional labeling efficiency. However, that is only true if the maximum cross-correlation achievable in the system is known. For that, a positive control for FCCS is required. In the same vein, the authors see about 70% cross-correlation for Sec8-GFP/Sec5-Halo. To evaluate what this corresponds to in actual interactions, the authors would need a positive control to determine what the maximum cross-correlation is (see point 1). The cross-correlation amplitude is limited by a multitude of parameters, including the overlap of the emission volumes of the two dyes, possible chromatic aberrations and shifts of the volumes, dark states and maturation/folding issues of the fluorescent proteins, and/or FRET. The authors comment on the non-correction for volume overlap on page 34. While that does not change comparison of relative fractions, it still means that absolute fractions cannot be determined. Thus, the 40% cross-correlation for Sec8-GFP/Sec5-Halo supports suboptimal labeling but cannot corroborate the absolute values. And the 70% cross-correlation for Sec8-GFP/Sec5-Halo, is likely and underestimate and could correspond to any absolute interaction from 70-100%.

We appreciate the reviewer's insightful comments regarding these matters, and have made appropriate modifications to the manuscript. For the first part, we determined HaloTag labeling efficiency using two independent methods: single molecule localization as described in page 20, and FCCS. These two approaches both estimate a 40% labeling efficiency, which gives us confidence in this result.

Regarding the positive dcFCS control in live cells, the highest cross-correlation amplitude measured in our setup was indeed in the range of 70-80%. Using the fraction of labeled Halo that was also Venus positive for the corrections, we obtain a 76% coupling. We have corrected our estimates for the exocyst subunit interactions based on this value, and assuming that Venus (YFP) and sfGFP folding efficiencies are similar, and we state this in the main text (page 11). We reason that this correction is a fair assumption as it is similar to a different positive control used in a recent paper from the Gradinaru lab, using the same dcFCS setup (Li et al., *Biophys. J.*, 2018). Figure 8 and page 14 now also reflect these corrections.

2. Page 12: The authors don't state the determined PSF size, nor the relevant size of the pixels. So one cannot evaluate whether the 4 pixel distance is justified (even if one tries to estimate the values from the given objective and camera). It would be easier for the reader if the data is provided.

This is an important point and we thank the reviewer for highlighting it. We have now included data for the experimentally determined point spread function of our TIRF objective with numerical aperture 1.49 in a new Supplementary Fig 7h. The experimentally determined lateral PSF = 244nm. The effective pixel size = 120nm/pixel. As is shown in Supplementary Fig7i, we had chosen a threshold of minimum 5-pixel distance separation to count molecules. As is evident this is >2 PSF and meets the Nyquist criterion.

3. Page 34, equation 1: The equation is correct if the brightness of both diffusing species is the same. If the brightness is not the same, the fluorophores contribute with the square of their brightness. Therefore, the statement on page 35 is not correct that $fD1+fD2=1$ if $fD1$ and $fD2$ are fluorescence contributions. $fD1+fD2=1$ holds only if $fD1$ and $fD2$ represent mole fractions. This, however, does not change the interpretation of the data as the authors see singly labeled species the brightness for both particles is the same.

To clarify, in our dcFCS model, the molecular brightness contribution is implicitly included in the “intensity fractions” $fD1$ and $fD2$ (Eq. 1, page 23). Under this definition, $fD1$ and $fD2$ do sum to 1. Also as the reviewer points out, since the complexes most likely each contain a single copy of the fluorescently labelled protein, $fD1$ and $fD2$ can also be interpreted as mole fractions. To make this clear we have now indicated this point on page 23.

REVIEWERS' COMMENTS:

Reviewer #1 (Remarks to the Author):

The authors have addressed my technical concerns. More generally, I am still of the opinion that work does not greatly advance our understanding of exocyst function, nonetheless the work appears to be very well executed and the conclusions are solid if not exciting.

Reviewer #2 (Remarks to the Author):

The paper is now suitable for publication in Nature Communications, in my opinion.

Signed,
Keith Mostov

Reviewer #3 (Remarks to the Author):

The authors have answered all queries.